# Robust coupling between the C-tactile afferent and the hair follicle in humans

Warren Moore[1] , Johan Nikesjö[2] , Otmane Bouchatta[2], Adarsh D. Makdani[3] , Pierre Hakizimana[2] ,
Mikael Rousson[4], Basil Duvernoy[2] , Sarah McIntyre[2] , Laura J. Pehkonen[2] , Anders Fridberger[2],
Francis McGlone[5], Hakan Olausson[2], Saad S. Nagi[2] and Andrew Marshall[1,2]

[1]*Department of Musculoskeletal Biology and Aging, University of Liverpool, Liverpool, UK*
[2]*Department of Biomedical and Clinical Sciences, Linköping University, Linköpin, Sweden*
[3]*Research Centre for Brain and Behaviour, Liverpool John Moores University, Liverpool, UK*
[4]*PioneerBio, Linköping, Sweden*
[5]*Department of Life Sciences, Manchester Metropolitan University, Manchester, United Kingdom*

Handling Editors: Nathan Schoppa & Vaughan Macefield

The peer review history is available in the Supporting Information section of this article (https://doi.org/10.1113/JP287706#support-information-section).

**Abstract figure legend** A single hair was carefully deflected using forceps during microneurography, which facilitates direct recording of single nerve fibre responses in awake participants. Recordings from 15 consecutive C-tactile (CT) afferents, where the response to hair deflection was investigated, showed that individual CT afferents respond to selective hair deflection (downward deflecting spikes in black trace). The results provide electrophysiological evidence that human CT afferents, similar to C-low-threshold-mechanoreceptors (C-LTMR) in mice, have a functionally relevant anatomical relationship to hair follicles.

**Warren Moore** MSc is a PhD candidate at the University of Liverpool, UK, and Linkoping University, Sweden. He earned a BSc in psychology and MSc in neuroscience at Liverpool John Moore's University. His PhD research area focuses on recording electrophysiology responses of cutaneous nerve afferents in the glabrous and hairy skin, using the technique of microneurography. He is particularly interested in the anatomical coupling between the C-tactile afferent and hair follicle, functional differences in low-threshold mechanoreceptor and nociceptor responses in fibromyalgia, and the ascending central pathway of touch and pain signals.

**Abstract**　C-low-threshold-mechanoreceptors (C-LTMR) were first discovered in the cat over 80 years ago and were subsequently documented in multiple mammalian species, including humans. Although their intimate association with hair follicles has been documented in mouse skin innervated by spinal ganglia, the functional anatomy in human skin is not known. The present study aimed to establish whether human C-LTMRs, also referred to as C-tactile (CT) afferents, have a functional association with hair follicles by determining their response to hair deflection using microneurography. Recordings from 15 consecutive CT afferents where the response to hair deflection was investigated showed that individual CT afferents respond to selective hair deflection. Mechanical hair plucking evoked after-discharge in CT afferents in the order of seconds following the stimulus. The results provide electrophysiological evidence that human CT afferents, similar to C-LTMRs in mice, have a functionally relevant anatomical relationship to hair follicles.

(Received 24 October 2024; accepted after revision 13 June 2025; first published online 23 July 2025)

**Corresponding author** A. Marshall: University of Liverpool, Department of Musculoskeletal Biology and Aging, Institute of Life Course and Medical Sciences, William Henry Duncan Building, West Derby Street, Liverpool, UK L7 8TX.　Email: andrew.marshall@liverpool.ac.uk

### Key points

- In humans, recordings from 15 C-tactile afferents where the response to hair deflection was investigated, showed that individual C-tactile afferents respond to selective hair deflection.
- Mechanical hair plucking evoked after-discharge in C-tactile afferents in the order of seconds following the stimulus.
- The results provide electrophysiological evidence that human C-tactile afferents, similar to previously reported C-tactile afferents in mice, have a functionally relevant anatomical relationship to hair follicles.

## Introduction

Slowly conducting C-fibres with low mechanical thresholds, C-low-threshold-mechanoreceptor (C-LTMR) afferents, were first discovered by Zotterman (1939) during the recording of action potentials from hairy skin supplied by the saphenous nerve in the cat in response to a stroking touch stimulation. Fifty years later, C-LTMRs were discovered in the human supraorbital nerve, an achievement that was only possible because of the development of an electrophysiological technique facilitating peripheral afferent activity recording, namely microneurography, that allows single afferent responses to be measured directly in awake humans (Nordin, 1990; Vallbo & Hagbarth et al., 1968). In humans, C-LTMRs are commonly termed C-tactile (CT) afferents (Olausson et al., 2010) and, similar to their non-human counterparts, are exquisitely sensitive to gentle touch (Löken et al., 2009; McGlone et al., 2014), with a mechanical threshold as low as 0.04 mN (Löken et al., 2009; McGlone et al., 2014; Watkins et al., 2017). They preferentially respond to gentle stroking touch at a velocity of $\sim$3 cm s$^{-1}$ (Löken et al., 2009), delivered at skin temperature (Ackerley et al., 2014). CT afferents also respond to focal mechano-cooling stimuli applied to the skin and can also respond to focal mechano-warming (Yu et al., 2024). In humans, CT targeted stimulation is positively correlated with an oxytocin release (Trotter et al., 2022; Walker et al., 2017), heart rate deceleration (Pawling et al., 2017) and pleasantness ratings (Essick et al., 2010) and, according to the social touch hypothesis (McGlone et al., 2014; Morrison et al., 2010), CTs have been proposed to play a fundamental role in pleasant, affiliative touch.

Human CT-afferents and non-human C-LTMRs have strong expression of the mechanosensitive ion channel PIEZO2 (Yu et al., 2024). Although this probably confers mechanical sensitivity on a molecular scale, the exquisite sensitivity of CT/C-LTMR afferents will also depend on the anatomical arrangement of the peripheral endings and could be engendered by functional coupling with hair follicles and/or peripheral endings in the epidermis. In mice, two potential C-LTMR subtypes have been identified: one expressing mas-related G protein-coupled receptor B4 (MrgprB4) and the other expressing vesicular glutamate transporter type 3 (VGlut3) and tyrosine hydroxylase (TH) (Li et al., 2011; Liu et al., 2007), both of which encircle hair follicles.

The mammalian hair follicle is situated within the dermis of the skin and, structurally, it comprises the

dermal papilla, inner and outer root sheath, and the hair shaft itself (Buffoli et al., 2014). In humans, single or multiple hairs may project from a hair follicle depending on its location (Lester & Venditti, 2007). At least three distinct types of hair follicles have been described in the hairy skin of the mouse (Li et al., 2011) and in the pig (Jiang et al., 2021): namely guard, awl/auchene and zigzag. Every hair follicle in human skin produces different types of hair over its lifetime: lanugo, vellus, intermediate and terminal hairs (Vogt & Blume-Peytavi, 2003). Lanugo hair is the soft, fine hair covering the intra-uterine fetus. Although sometimes seen in newborns, under normal circumstances, lanugo hair disappears within a few weeks of birth (Buffoli et al., 2014; Gworys & Domagala, 2003). It is the innervation of vellus and terminal hairs that is the focus of this report. There are no hair follicles in glabrous skin in humans (McGlone & Reilly, 2010), although a small number are present in the middle of the mouse hind-paw (Walcher et al., 2018).

The molecular characterization of C-LTMRs in the mouse has facilitated visualization of the peripheral endings of C-LTMRs. Indeed, elegant genetic labelling studies describe a distinct combination of murine low-threshold mechanoreceptors innervating the three types of hair follicle, namely: guard ($A\beta$ rapidly (RA) adapting low-threshold mechanoreceptor (LTMR) afferents), zigzag ($A\delta$ and TH+ C-LTMR) and awl (TH+ C-LTMR, $A\delta$, $A\beta$ RA) (Li et al., 2011). $A\beta$-LTMRs, $A\delta$-LTMRs and TH$^+$ C-LTMRs terminate with longitudinal lanceolate endings that are associated with the hair follicles (Li et al., 2011; Takahashi-Iwanaga, 2000). This anatomical coupling of C-LTMR afferents with the hair follicle probably facilitates the exquisite sensitivity of TH$^+$ C-LTMRs to hair deflection (Abraira & Ginty. 2013; Li et al., 2011). Presumptive C-LTMRs expressing MrgprB4 also encircle hair follicles (Liu et al., 2007). However, human CT-afferents do not express either TH or MrgprB4 (Yu et al., 2024) and direct anatomical evidence of the location of their endings in the skin is lacking.

Movement of the hair shaft activates $A\beta$ and $A\delta$ fibres in both the cat and rabbit, indicating a close proximity of the nerves to the hair follicle (Brown & Iggo, 1967; Burgess et al., 1968). In cats, Iggo (1960) also showed that some, but not all, C-LTMRs are sensitive to hair movement. Indeed, movements of both guard and down hairs as small as 10–20 μm are sufficient to induce firing (Iggo & Kornhuber, 1977). It is suggested that small movements of hairs could contribute to after-discharges in C-LTMRs (Iggo & Kornhuber, 1977) as a result of restorative movements of the epidermal surface following skin displacement.

Data on the responsiveness of CTs to hair movement in humans are sparse. Nordin (1990) described a single CT unit innervating human scalp hair that responds vigorously to displacement and replacement of a single hair, but not to sustained displacement. However, in the study by Ackerley et al. (2014), CTs dis not respond to air puffs, which activated $A\beta$ hair follicle afferents. *In vitro* mechanical stimulation of human scalp root sheath (hair follicle) cells, mimicking hair deflection, resulted in putative C-LTMR depolarization, indicating an anatomical coupling between C-LTMRs and the hair follicle (Agramunt et al., 2023). Furthermore, serotonin and histamine released by the outer root sheath cells resulted in the firing of co-cultured murine sensory neurons, detected by calcium imaging, thus suggesting a chemical signalling mechanism. However, Agramunt et al. (2023) identified putative scalp C-LTMRs by the presence of TH, which is not expressed in CTs from human spinal dorsal root ganglia (DRG) (Yu et al., 2024). In the absence of a molecular marker that reliably stains CT afferents in human hairy skin to determine the precise anatomical relationship with hair follicles, evidence for the anatomical coupling is reliant upon *in vivo* experiments. Accordingly, we used the single-unit electrophysiological recording technique of microneurography to measure CT firing responses to hair stimulation. We hypothesized that (1) CTs will respond to selective hair deflection; (2) CT firing frequency during hair deflection will not be substantially different to soft slow brushing; and (3) CTs will respond to mechanical hair removal.

## Methods

### Ethical statement

Ethical approval was given by the ethics committees of Linköping University (dnr 2020–0 4426), Swedish Ethical Review Authority (Dnr 2024-0 5844-01) and Liverpool John Moores University (14/NSP/039). This study conformed to the *Declaration of Helsinki* and written consent was obtained from each participant.

### Microneurography

**Microneurography participants.** Sixteen participants (10 males and six females) with a mean age of 30.7 years (range 19–53 years) were recruited from Linkoping University, Sweden, and Liverpool John Moores University, Liverpool, UK. The inclusion criterion for this study comprised healthy participants aged >18 years. The exclusion criteria for this study comprised neurological or musculoskeletal disorders, skin diseases (e.g. psoriasis), diabetes and pain-relieving or psychoactive medication. Participants were compensated for their time.

**Microneurography experimental setup.** An epoxy-resin insulated 20–50 mm long tungsten recording electrode (2 and 5 MΩ, 200 μm shank diameter) was inserted

percutaneously, under ultrasound guidance (LOGIQ P9; GE Healthcare, Chicago, IL, USA) towards the target nerve (12× radial, 5× posterior/dorsal antebrachial and 2× superficial peroneal). Neural activity was sampled at 20 kHz and recorded using a data acquisition system (LabChart software, version 8.1.24, and PowerLab 16/35 hardware PL3516/P; ADInstruments, Oxford, UK). Single action potentials were identified semi-automatically, with visual verification on an expanded time scale. Furthermore, repeat trials for each stimulus were conducted to ensure reproducibility. Nerve fibres were classified by brush sensitivity to soft artist/goat hair and rough synthetic brushes ∼4 cm wide, and Semmes-Weinstein monofilaments (nylon fibre; Aesthesio, Bioseb, Pinellas Park, FL, USA). Monofilaments (0.08–3000 mN) were used to determine adaptation type and mechanical threshold. A manual squeezable air puff device was used to deliver a weak but focused jet of air over the receptive field. Data were collected from all sixteen participants. Fifteen CT units and three $A\beta$ hair follicle afferents (HFA) were recorded.

**Microneurography analysis.** Captured neural data were analysed using LabChart, version 8.1.24 (ADInstruments), where threshold crossing was used to distinguish action potentials from noise with spike morphology confirmed by template matching (spike height, width and shape) to action potential shape during soft brushing trials. For each stimulus, spike count, mean instantaneous frequency (calculated between the first and last spike) and peak instantaneous frequency were calculated. Recordings were discarded if multiple units were present (e.g. non-physiological spike intervals/firing rates) or if spike amplitudes were not distinct from the noise, preventing secure action potential identification. The neural response was evoked (or modulated) only when the specific area of skin (the receptive field) was stimulated. RStudio, version 2023.06.2 (Posit, Boston, MA, USS) running R, version 4.3.0, was used to carry out statistical analyses for the neural data. A repeated measures *t* test compared mean instantaneous frequency between brushing and hair deflection. For each unit, a single mean response was calculated based on multiple stimulus trials. The plotted data represent the mean response for each unit.

**Microneurography unit classification.** Nerve fibres were first classified as $A\beta$-fibre or C-fibre, indicated by upwards ($>30$ m s$^{-1}$ conduction velocity) and downwards ($<2$ m s$^{-1}$ conduction velocity) deflecting spikes, respectively, measured using electrical stimulation. A delayed response in the order of hundreds of milliseconds was a defining characteristic of C fibres. The unit was classified as an LTMR (A or C) if it responded to slow soft brushing (delivered manually by a trained experimenter with a soft brush) at a velocity ∼3 cm s$^{-1}$, and had a mechanical threshold below of 4 mN or below, defined as at least a single spike response on at least 50% of occasions to monofilament indentation'. Response to a supra-threshold monofilament force was first recorded, followed by systematically decreasing the force until no response is recorded. Then, the mechanical threshold was confirmed by re-testing the lowest force eliciting a response. Adaptation type was classified as either slow or rapidly adapting, using 100 mN von Frey hold. Among $A\beta$ RA-LTMRs, units that responded to hair deflection were identified as HFAs as per criteria used by Vallbo et al. (1995). Following classification as an HFA or CT unit, the hair deflection and plucking protocol were followed.

**Microneurography hair deflection.** Hairs in the receptive field of a unit were carefully deflected or plucked using forceps under magnification, ensuring no contact with the skin or using a custom-built device consisting of a small mechanical clip attached by a flexible wire. Each hair was deflected against the direction of hair growth in that deflection in the opposite direction was not possible because of the risk of inadvertent direct stimulation of the skin.

A minimum of two spikes was chosen to provide an extra layer of confidence that the response seen was an evoked one and also because this is needed to detect an instantaneous firing frequency. When possible, the firing response to deflection of more than one individual hair was studied. Following multiple (between two and six) repetitions of hair deflection, the hair was pulled with increasing force until removed. The presence of spontaneous activity in response to hair removal was documented.

## Hair deflection tracking

To calculate hair displacement velocity and hair displacement induced skin movement, an optical flow computation experiment was included. The approach uses the SAM2 (Segment Anything Model 2) (Ravi et al., 2024) for initial image segmentation, focusing on the hair deflection device. Post-processing steps include refining the segmentation mask by applying morphological operations, followed by contour extraction and simplification. The tip of the device is identified as the point of highest curvature along the main contour, with a constraint to search within a defined radius of the previous frame's tip location to ensure temporal consistency. The tip's position is tracked across frames, and its speed is calculated and smoothed using a Gaussian filter (Fig. 1). This method allows for robust object segmentation and tip tracking in video sequences, providing quantitative data

on the object's movement dynamics (see also Supporting material Video S1).

**Hair deflection video acquisition and preprocessing.** High-resolution videos ('.mov' file) (1920 × 1080 pixels) were recorded at 60 frames s$^{-1}$ using an iPhone 15 Pro (Apple Inc., Cupertino, CA, USA) at 5× optical zoom. The iPhone was stabilized on a standard photography tripod equipped with a smartphone mount. The tripod's height, distance and angle were adjusted to centre the field of view on both the deflected hair strands and the forceps tip, maintaining these features in clear view throughout the recording. Once the setup was established, no camera repositioning was required. Multiple video recordings were acquired on the forearms of two participants. The resulting footage provided optimal contrast between dark hair strands/forceps tip and the lighter skin background, enabling precise optical flow computation. Pixel calibration on a subset of video recordings established a resolution of 465 pixels cm$^{-1}$.

**Optical flow computation.** Optical flow fields were computed using the dense Farnebäck algorithm (Farnebäck, 2003) via OpenCV's cv2.calcOpticalFlowFarneback (Bradski, 2000). Parameters included a pyramid scale factor (pyr_scale = 0.6), four pyramid levels, a window size of 7 pixels, up to eight iterations per level, and a polynomial expansion with poly_$n$ = 5 and poly_sigma = 1.5. These settings balanced sensitivity to subtle, localized displacements against noise robustness, capturing a practical range of motion magnitudes, from small-scale surface shifts to larger hair deflections and forceps tip displacements.

**Computation of cumulative displacement fields.** To quantify net motion over time, incremental flow vectors computed between consecutive frames were summed cumulatively at each pixel location. Thus, the arrays cum_flow_x and cum_flow_y represented total displacement from the start of recording to any given frame. By the end of the video, these cumulative fields described how far and in what direction each point had moved from its initial position.

**Visualization and scaling.** Two outputs were produced from the cumulative flow data:

(1) *Final Still Image (Quiver Plot).* After processing the entire video, a quiver plot of the cumulative displacement field was generated. Arrows indicated direction and magnitude of total displacement, and a global colour scale (jet colormap) was applied, normalized to the maximum cumulative flow observed. This global normalization highlighted even subtle net movements accrued over time, providing a clear, end-state visualization of all displacement magnitudes.

(2) *Time-Resolved Video (Green Vectors).* The video depicts instantaneous optical flow vectors (shown in green) superimposed on the original video frames, with a timestamp displayed in the top-left corner. Small green arrows indicate direction and magnitude of local motion, updating frame-by-frame as the forceps deflect individual hair strands against the skin background. The high contrast between the dark hair strands, silver forceps tip, and light skin enables precise motion tracking. The vector field reveals both the forceps movement and the associated

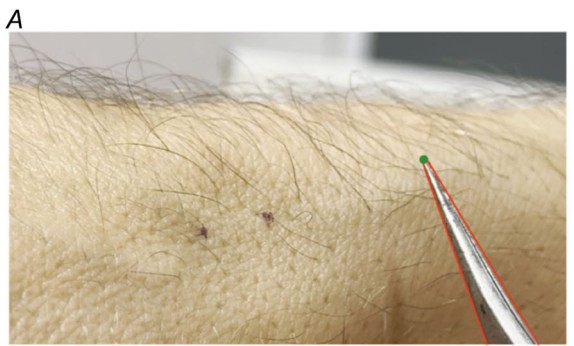
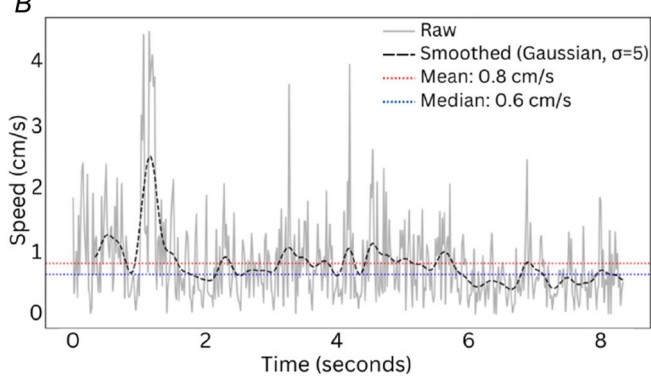

**Figure 1. Motion tracking of forceps' tip during hair deflection**
*A*, a single frame from the video (see Supplemental Video 1) shows the convex hull contour (red dotted line) around the forceps, with the highest curvature point marked (green dot) at the tip, near the contact point between the forceps and the hair. The SAM2 model was used to track this point in real-time, effectively capturing the movement and speed of the hair during stimulation (mean = 0.8 cm s$^{-1}$). *B*, the speed profile of the forceps' tip over time, displaying raw speed data (grey) and smoothed data (black dashed line, $\sigma$ = 5), shows variations in movement speed that correspond to the interaction between the forceps and the hair. [Colour figure can be viewed at wileyonlinelibrary.com]

hair displacement at the same time as demonstrating minimal perturbation of surrounding tissue (see Supporting information, Video S2).

**Software implementation.** All analyses were conducted in Python, version 3.11.5 (https://www.python.org) using OpenCV (Bradski, 2000), NumPy (Harris et al., 2020), Matplotlib (Hunter, 2007) and tqdm (da Costa-Luis, 2019) for progress reporting.

## Optical coherence tomography (OCT)

**Experimental setup.** A single terminal hair located on the right forearm of a participant was selected for this additional study. The hairs within a 1 cm perimeter around the selected terminal hair were shaved to limit deflection to only the selected hair. The participant was positioned comfortably in a medical chair with pillows to support the forearm, minimizing static loads on muscles and joints, and ensuring stability throughout the 2 h experiment.

The experiment involved five different types of skin deformation generated by five different actuation systems, as well as a high spatiotemporal resolution camera to measure the skin deformations. The five stimuli were: hair deflection using a custom hook, as well as four indentations using Semmes-Weinstein monofilaments (Aesthesio Precise Tactile Sensory Evaluator Kit; DanMic Global, San Jose, CA, USA). The forces delivered by the monofilaments cover the range of threshold values for the C-LTMRs recorded in the main study (0.4, 0.7 and 4 mN), as well as a larger force well above threshold (6 mN). Both the hook and the monofilaments were mounted on a linear motor (LM1247-040; Faulhaber, Schönaich, Germany) to ensure repeatability and facilitate comparison between trials. For the hair deflection, the hook's motion direction was opposed to the hair's natural bending, with a speed set to 10 mm s$^{-1}$ and a total stroke of 3 cm, which was sufficient for the hair to reach its maximum bending and release itself from the hook to return to its resting state. The monofilaments indented the skin for 500 ms. These movement parameters were chosen to resemble how stimulations were conducted in the main study. Skin deformation was captured using an OCT system (TEL3201C; ThorLabs, Newton, NJ, USA; central wavelength 1300 nm). OCT is an interferometric imaging technique capable of generating depth-resolved images (∼5.5 µm resolution in the air). In a specific custom mode, it is possible to trade the large, millimetres wide, structural image and slow refresh rate (few Hz), which is the standard of OCT systems, to a very narrow image (one single A-line) at a high refresh rate (10 kHz in our case). This unique setting allows tracking highly dynamic changes of the skin surface over 3 s of recording.

Skin deformations were measured at nine single locations around the hair, forming a 3 × 3 grid of 2 × 2 mm. Five trials per condition were recorded at each location, totalling 45 trials per condition. The skin at its resting state (idle, no stimulation) was also recorded similarly to the other conditions.

**Data processing.** Trials containing artefacts (high heartbeat, participant motion, hook or hair or monofilament crossing the OCT light beam) that obscured the active deformation were discarded (23% of all trials, highest rejection being on 4 mN). A segmentation algorithm was applied to track the skin surface location relative to the OCT probe over time for each trial. For the indentation stimuli, we measured the skin location at the resting state before contact and after contact between the monofilament and the skin. Two windows of 200 ms, before and after this event, were selected. Within these windows, skin location was averaged and then subtracted to obtain the amount of indentation for each trial.

Regarding the hair deflection condition, there was no statistical skin deformation during the hair deflection. However, to compare it with the indentation conditions, we focused on the moment when the hair was at its maximum bending position and released itself from the hook to return to its resting state. It is reasonable to hypothesize that the maximal skin displacement occurs when the hair is at its maximum deflection. Hence, similarly to the indentation conditions, we measured the difference in skin surface location before and after the release of the hair from the hook. For the idle (resting) condition, two consecutive windows of 500 ms were taken and subtracted. The longer window length was to compensate for potential biological motions.

Finally, for each stimulus type, the 10 highest calculated displacements were selected. The reason is that the positioning of the different stimulus instruments on the skin was performed manually, which means that the contacts made by the monofilaments were located at slightly different locations on the skin. Therefore, for any given stimulation, some of the nine measured locations may have been more optimally located to capture skin displacement than others, but this could not be determined ahead of time.

## Results

### Unit characterization

Fifteen consecutive CT units with visible hair in the receptive field were recorded (locations shown in Fig. 2*A*). Recordings were taken between September 2021 and February 2023, during which CT afferents that did not respond to hair deflection or CT afferents without a visible hair within the receptive field were not seen. Various

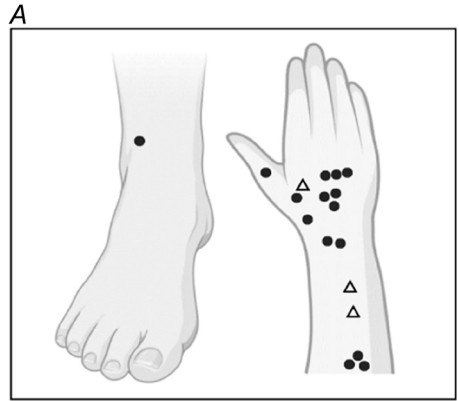

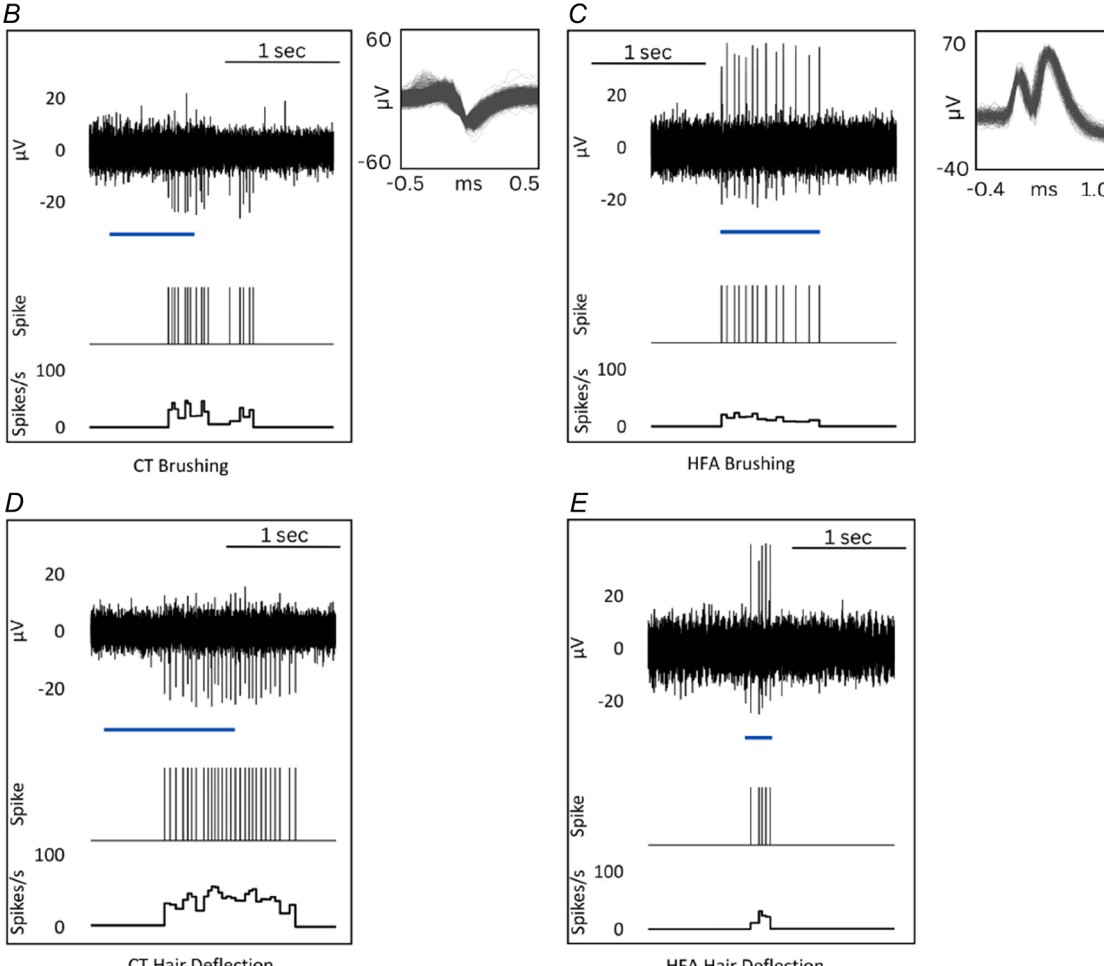

**Figure 2. CT and HFA response to brushing and hair deflection**
*A*, location of receptive fields for CT (dots) and HFA (triangles) units. For (*B*) to (*E*), raw neural traces are shown above spike markers and instantaneous frequency. Left: CT response to (*B*) brushing and (*D*) hair deflection is shown for the same unit. Right: HFA responses to (*C*) brushing and (*E*) hair deflection is shown for the same unit. Superimposed individual spikes are shown on an expanded timeline in the smaller boxes for both units. Blue bars indicate the approximate stimulus duration. [Colour figure can be viewed at wileyonlinelibrary.com]

features of 12 of these CTs have been previously reported previously (Bouchatta et al., 2023; Yu et al., 2024), but the hair movement responses are described here for the first time. All CTs were characterized based on a mechanical threshold of 4 mN or less (mean = 0.92 mN, SD = 0.87 mN, minimum = 0.4 mN, maximum = 4.0 mN, $n = 15$) or a noticeable delay in spiking after skin stimulation, conduction velocities within the C-fibre range (mean = 1.09 m s$^{-1}$, SD = 0.31 m s$^{-1}$, $n = 3$) and sensitivity to soft brushing (instantaneous frequency: mean = 21 spikes s$^{-1}$, SD = 6 spikes s$^{-1}$; peak frequency: mean = 44 spikes s$^{-1}$, SD = 13 spikes s$^{-1}$; $n = 15$) (Figs 2 and 3).

All HFAs ($n = 3$) were soft brush sensitive (mean instantaneous frequency: mean = 31 spikes s$^{-1}$, SD = 15 spikes s$^{-1}$; peak frequency: mean = 70 spikes s$^{-1}$, SD = 118 spikes s$^{-1}$) and responded to air puff delivered via a hand-held manual squeezable device or by blowing on the skin (mean instantaneous frequency: mean = 26 spikes s$^{-1}$, SD = 35 spikes s$^{-1}$; peak frequency: mean = 63 spikes s$^{-1}$, SD = 82 spikes s$^{-1}$) and manual hair deflection (mean instantaneous frequency: mean = 12 spikes s$^{-1}$, SD = 7 spikes s$^{-1}$; peak frequency: mean = 93 spikes s$^{-1}$, SD = 95 spikes s$^{-1}$) (Figs 2 and 3). The conduction velocity of a single HFA unit was recorded electrically at 34.9 m s$^{-1}$.

### Hair deflection response of CTs

To elucidate the functional coupling between the hair follicle and CT, responses to hair deflection were recorded. All CT units ($n = 15$) responded to slow manual deflection of the hair (mean instantaneous frequency: mean = 12 spikes s$^{-1}$, SD = 5 spikes s$^{-1}$; peak frequency: mean = 45 spikes s$^{-1}$, SD = 20 spikes s$^{-1}$), with responses comparable to brushing (Figs 2 and 3). CT afferents fired in response to hair deflection of more than one hair in the receptive field, as tested separately (Fig. 4). They also responded to air puff stimulation (mean instantaneous frequency: mean = 10 spikes s$^{-1}$, SD = 13 spikes s$^{-1}$; peak frequency: mean = 30, SD = 31 spikes s$^{-1}$).

Hair deflection response of a single CT innervated by two individual hairs was recorded. Both hair one (mean instantaneous frequency = 7 spikes s$^{-1}$ and peak instantaneous frequency = 13 spikes s$^{-1}$) and hair two (mean instantaneous frequency = 23 spikes s$^{-1}$ and peak instantaneous frequency = 40 spikes s$^{-1}$) displayed hair deflection sensitivity (Fig. 4 ).

### Hair plucking after-discharge

After-discharge responses of six CTs were recorded following hair plucking (example shown in Fig. 5). All

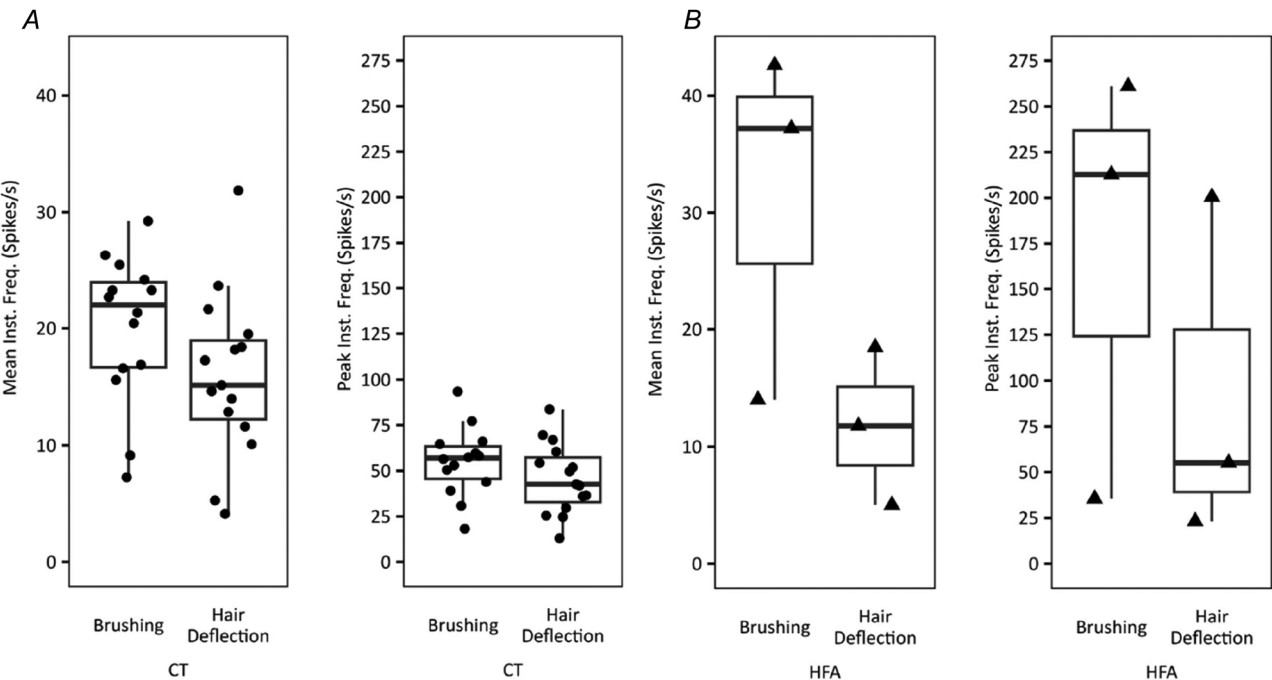

**Figure 3. CT and HFA responses to brushing and hair deflection**
Data points represent mean for individual units and bold lines represent the median value. *A*, CT mean instantaneous frequency and peak instantaneous frequency response to brushing and hair deflection. The response to brushing and hair deflection was statistically indistinguishable (Mann–Whitney test: mean frequency, *P* = 0.057; peak frequency: *P* = 0.104). *B*, HFA mean instantaneous frequency and peak instantaneous frequency response to brushing and hair deflection. The response to brushing and hair deflection was statistically indistinguishable (Mann–Whitney test: mean frequency, *P* = 0.250; peak frequency: *P* = 0.250.

six units exhibited after-discharge immediately following mechanical hair removal, lasting for a mean of 7.1 s (SD = 3.2 s, minimum = 2.3 s, maximum = 11.7 s), with a mean instantaneous frequency of 5 spikes s$^{-1}$ (SD = 5.0 spikes s$^{-1}$) and peak instantaneous frequency of 26 spikes s$^{-1}$ (SD = 15 spikes s$^{-1}$). CTs remained mechanically responsive after removal of individual hairs. HFA units ($n = 2$) did not exhibit after-discharge following hair plucking (mean instantaneous frequency: mean = 0 spikes s$^{-1}$, SD = 0 spikes s$^{-1}$; peak instantaneous frequency: mean = 0 spikes s$^{-1}$, SD = 0 spikes s$^{-1}$).

In a single participant, spontaneous generalized piloerection was observed during a stable recording of a CT afferent, which was associated with a rapid burst of spikes (Fig 6).

### Hair deflection induced skin displacement

**OCT.** If the CT afferents were not located in close proximity to hair follicles but rather closer to the skin surface similar to other types of tactile mechanoreceptors, then, when the deflected hair is near its maximum bending or is being released, and when the units fire the most (for examples, see Fig. 3, blue bars), the skin surface deformation should be comparable to the mechanical stimuli used to assess the units' threshold (monofilaments between 0.4 and 4.0 mN). To test this hypothesis, we conducted an additional experiment to quantify the amount of surface skin displacement during monofilament indentations, hair deflection and when the skin is at rest. A pairwise comparison (Tukey's honestly significant difference test, $P$ value adjusted with a family-wise alpha of 0.05) was conducted to compare skin displacements among the different conditions (Fig. 7). Significant differences ($P < 0.05$) were found among all conditions except between the idle and hair deflection conditions ($P < 0.25$). The average displacement (in microns) for each condition was: idle at 2.04 [95% confidence interval (CI) = 0.90–3.17]; hair deflection at 8.14 (95% CI = 5.48–10.80); monofilament 0.4 mN at 17.09 (95% CI = 13.63–20.54); monofilament 0.7 mN at 25.28 (95% CI = 20.95–29.60); monofilament 4.0 mN at 60.93 (95% CI = 56.39–65.47); and monofilament 6.0 mN at 79.18 (95% CI = 71.70–86.67). This suggests that different types of stimulation had a different impact on the skin surface and that bending one hair induced less surface skin deformation than a 0.4 mN monofilament indentation. Furthermore, no significant difference was detected in surface skin displacement between the skin at rest (no stimulation) and during the transition of hair from maximal bending to its natural position. We have shown that any such deformation is significantly smaller than that caused by even the smallest monofilaments.

**Cumulative flow vector analysis.** In addition to the OCT findings described above, we employed a cumulative flow vector analysis of high-resolution video recordings to further assess local skin movement during hair deflection. This approach allowed us to determine whether CT afferent responses are driven directly by hair movement rather than by any secondary skin deformation. As shown

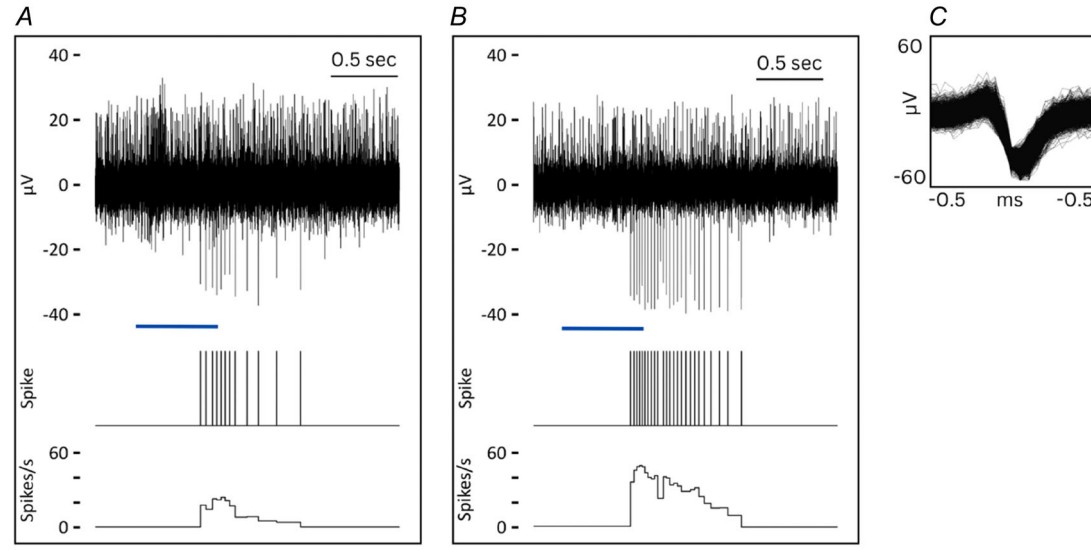

**Figure 4. CT response to deflection of multiple hairs**
Individual hair deflection responses of a single CT activated by deflection of two visible hairs in the receptive field. *A* and *B*, raw neural trace, spikes and mean instantaneous frequency response to two individual hairs innovating the same receptive field of a single CT. *C*, superimposed individual spikes are shown on an expanded timeline in the smaller boxes for both units. Blue bars indicate the approximate stimulus duration. [Colour figure can be viewed at wileyonlinelibrary.com]

in Fig. 8, which is representative of four video recordings from two different individuals, the Farnebäck optical flow algorithm detected a sharply localized displacement field corresponding to the forceps' trajectory and the associated hair deflection, whereas the surrounding skin remained essentially static. The associated time-resolved video (see Supporting information, Video S2) dynamically illustrates instantaneous optical flow vectors superimposed on the video frames, capturing both the forceps movement and the hair displacement with minimal perturbation of adjacent tissue. Together, these converging results confirm that the mechanical stimulus is confined to the hair, reinforcing the interpretation that CT afferent activation is primarily as a result of direct hair movement.

## Discussion

In the present study, we used microneurography to record CT firing responses to hair deflection and their after-discharge following mechanical hair removal (plucking). All CTs evoked a response to hair deflection, which was similar in magnitude to brushing. CTs also displayed after-discharge immediately following hair plucking. The response of CTs was similar for both slow manual hair deflection and previously documented pre-ferred stimulation (slow gentle brushing), indicating that a very focal stimulus (hair deflection) is a strong driver of action potentials in this fibre type.

Currently, there is no specific immunohistochemical marker to selectively stain for CTs in human skin. However, the findings reported in the present study provide evidence, in line with findings in mice (Li et al., 2011), showing that the CT has a close and functional anatomical association with the hair follicle in humans. A direct comparison of hair deflection responses is difficult because of differences between human hair and animal fur. In mice, C-LTMRs, which form lanceolate endings around awl/auchene and zigzag hair follicles, express TH and VGLUT3, and TH is used as a marker for this population of fibres (Li et al., 2011). Although Agramunt et al. (2023) identified TH positive neurons forming lanceolate endings around human scalp hair, TH has not been found to be expressed in putative C-LTMRs using transcriptional profiling of human spinal DRG neurons (Yu et al., 2024). Therefore, it is unclear whether TH

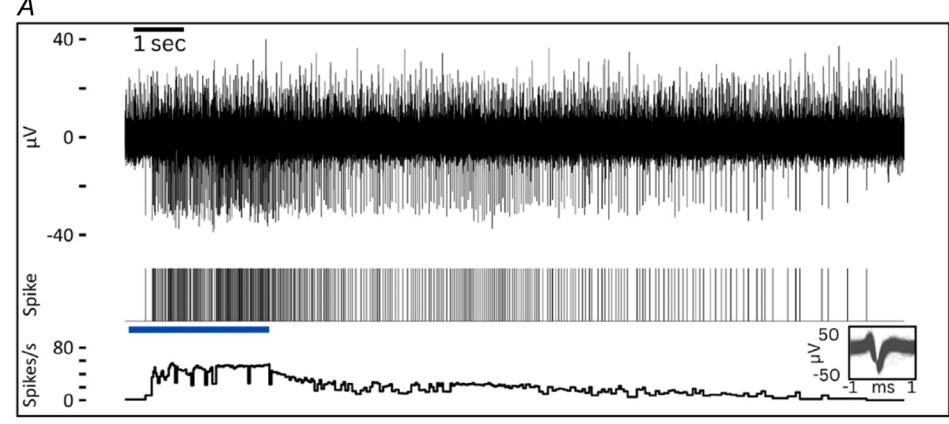
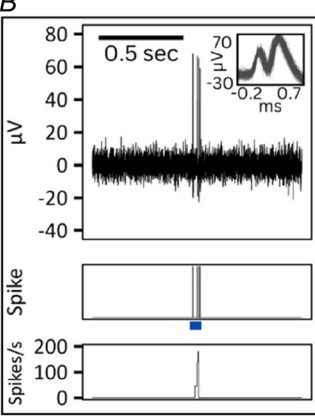

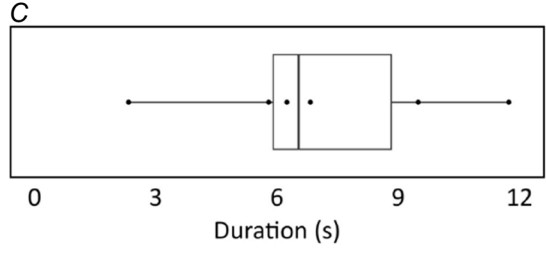

**Figure 5. Hair plucking after-discharge response**
*A*, CT firing response (top) and instantaneous frequency (bottom) following hair plucking. The blue bar indicates hair pulling duration with pulling force increasing until the hair was plucked out at the end of the stimulus. An after-discharge was observed following hair plucking. Individual spikes are superimposed on an expanded time scale in the smaller box. *B*, HFA after-neural response (top) and instantaneous frequency (bottom) following hair plucking. The blue bar indicates hair pulling duration with pulling force increasing until the hair was plucked out at the end of the stimulus. No after-discharge was observed. *C*, boxplot showing duration of after-discharge following hair plucking for each unit (dots). Median = 6.5 s (interquartile range = 2.9) after-discharge is represented by the vertical line and mean duration was 7.1 s (SD = 3.2 s). [Colour figure can be viewed at wileyonlinelibrary.com]

positive fibres described by Agramunt et al. (2023) in the human scalp represent CT/C-LTMR afferents or whether CT afferents in the trigeminal region are molecularly distinct from CT afferents that innervate skin supplied by spinal ganglia.

The proposed close anatomical relationship between CTs and hair movements can adequately explain their exquisite sensitivity to mechanical stimuli. Indeed, force activation thresholds of CTs using monofilaments should be interpreted with caution given that hairs may be inadvertently moved during skin stimulation. Data on the responsiveness of CTs to hair movement in humans are sparse, and we can only speculate on why our results indicate a greater CT sensitivity to hair deflection compared to previous literature. We undertook a rigorous systematic exploration of 15 consecutively recorded CTs where a single hair was carefully deflected under magnification, and perhaps previous studies have not focused on hair deflection responses in detail. Nordin (1990) described a single CT unit innervating human scalp hair that responds vigorously to displacement and replacement of a single hair, but not to sustained displacement. Ackerley et al. (2014) stated that CTs do not respond to air puffs, which activated Aβ hair follicle afferents. Given that CTs can innervate more than one follicle within a receptive field, it could also explain the

receptive field properties of CTs which typically exhibit several hotspots of mechanical sensitivity, which could potentially reflect the location of hair follicles (Wessberg et al., 2003). CTs remained sensitive to soft brushing and/or vonFrey monofilament indentation following the removal of individual hairs. However, it is not possible to know whether these were responses to direct skin stimulation or the result of the movement of other hairs within the receptive field. We did not assess whether removal of hairs changed the receptive field properties (e.g. altered the number or sensitivity of the hotspots after shaving of the hair as described by Wessberg et al. (2003). To what extent CTs/C-LTMRs have a role in the perception of individual hair deflections is unknown. Hair follicles in humans are also co-innervated by Aβ LTMR afferents. Unlike CTs/C-LTMRs, which have slow velocities and elicit poorly localized perceptions (Olausson et al., 2008), Aβ hair follicle and Aδ (D-hair) LTMR afferents have fast conduction velocities and other properties in keeping with a discriminative role. In rodents, D-hair afferents show a robust response to vibration (Furukawa et al., 2009; Lechner & Lewin, 2013) and, in both humans (Aβ hair follicle afferent LTMRs) and rodents (Aβ hair follicle afferent LTMRs and D-hair afferents), have firing frequencies that show a linear relationship to brushing velocity (Bai et al.,

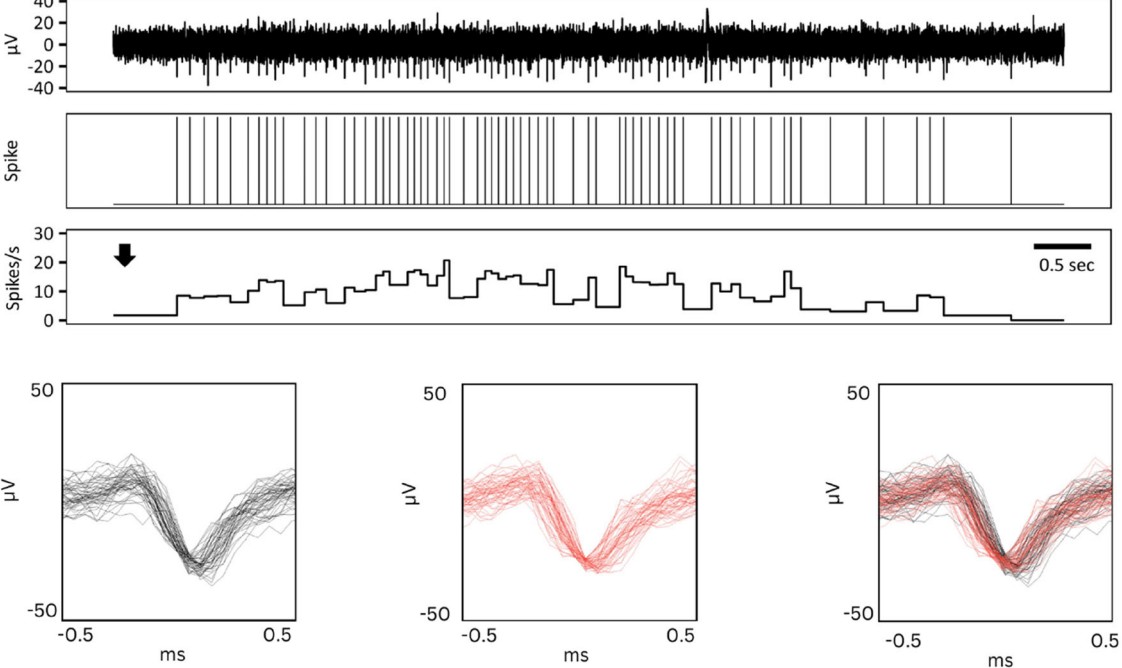

**Figure 6. Spontaneous piloerection**
Spike markers (top) and instantaneous firing frequency (centre) during spontaneous piloerection. In a single participant, a generalized spontaneous piloerection was observed, which coincided with the participant waking up from drowsiness. The approximate onset of the 'goosebumps' is indicated by the arrow. The boxes on the bottom show superimposed individual spikes on an expanded time scale for the unit activated during the piloerection superimposed during a preceding mechanical stimulus (hair deflection) (left), during piloerection (centre), and both overlaid (right). [Colour figure can be viewed at wileyonlinelibrary.com]

2015; Greenspan et al., 1992; Löken et al., 2009). A role for CTs in mechanically evoked itch delivered using vibratory stimulation of vellus hairs has been postulated in humans (Fukuoka et al., 2013) for which a prerequisite would be a close anatomical relationship between CTs and hair follicles. However, mechanically evoked itch on hair vibration is only reliably evoked in trigeminal innervated skin. In support of an anatomical association, a weak positive relationship between hair follicle density and affective touch pleasantness has been described (Jönsson et al., 2017). Synchronous body-wide activation of CT afferents during an episode of generalized piloerection (Fig. 6) is probably relevant to the tingling sensation that accompanies goosebumps (Harrison & Loui, 2014). Piloerection can occur under a variety of circumstances including as a response to emotional stimuli of positive or negative valence (Harrison & Loui, 2014; McPhetres & Zickfield, 2022; Tihanyi et al., 2018). Occurring as a psychophysiological response to a rewarding stimulus, goosebump-induced mass CT afferent activation may contribute to the positively-valanced affective state variously termed a frisson, aesthetic chills, psychogenic chills, or, given the pleasurable nature of the skin tingling, a skin orgasm.

Our findings suggest there is a close functional anatomical relationship between CT afferents and hair follicles, but there are several possible mechanisms by which hair deflection could induce firing. It could be argued that, during the process of hair deflection, other hairs were also moved causing a brushing against the skin surface. However, hair deflecting stimuli were delivered under magnification and great care was taken to avoid either direct or indirect mechanical stimulation of the skin surface. Therefore, technical factors such as this appear very improbable. One possibility is that hair movement causes deformation of the peri-follicular skin, and it is this that initiates firing of CTs. Only two (of 15) CTs had a mechanical threshold as low as 0.4 mN, which our OCT experiment confirmed produces 17 μm skin displacement. Given that hair deflection induced skin displacement (8 μm) was significantly less than this (8 μm, which is around half the displacement for that of a 0.4 mN monofilament), it is reasonable to suggest that CTs were not responding to hair deflection induced skin displacement, but rather to the hair deflection stimulus itself. However, it remains possible that peri-follicular skin movement, when present, may also contribute. Comparison with field units is interesting in this respect. Field units in humans have exquisite mechanical sensitivity (Nagi et al., 2019) and do not respond to hair deflection. Although the peripheral endings of field units in humans are yet to be defined, field afferents in mice, which have expansive receptive fields, are intimately related to hair follicles yet they do not fire in response to hair deflection (Bai et al., 2015). This indicates that having an intimate anatomical arrangement with hair follicles does not necessarily imply that these endings fire in response to hair deflection.

Hair movements and bending will cause mechanical strain within the follicle. To induce firing of CT afferents, this would require the peripheral endings of CT afferents

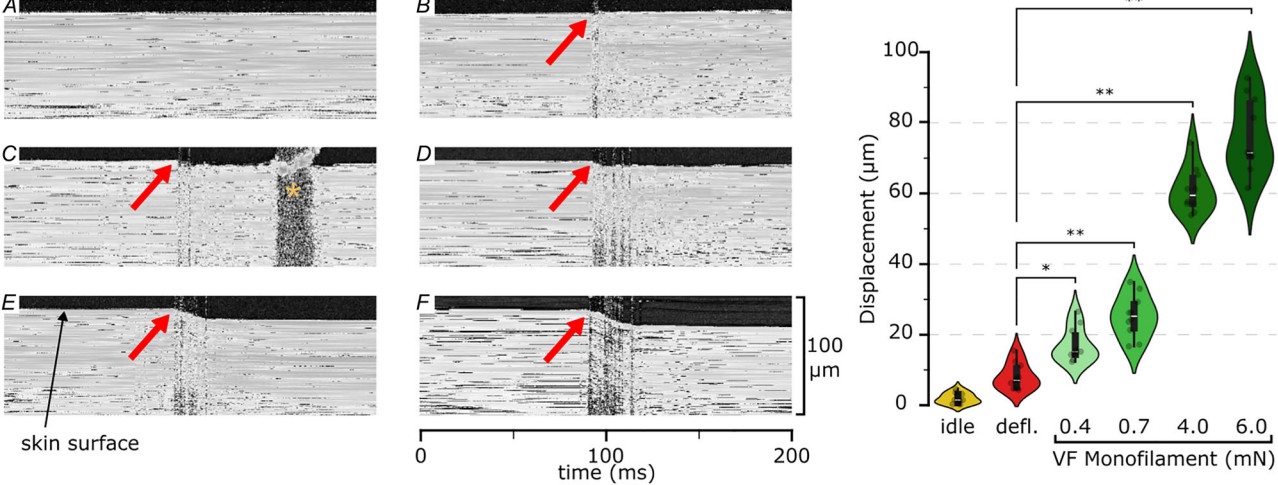

**Figure 7. Optical coherence tomography**
The left images display one-dimensional videos (A-line over time) of *in vivo* OCT recordings of hairy skin. The skin is depicted in white, illustrating both its surface and subsurface tissues, black represents the surrounding air. The skin motion is shown under six conditions: (*A*) idle (no stimulation); (*B*) hair release after complete deflection; (*C*, *D*, *E* and *F*) impact of Semmes-Weinstein monofilament (0.4, 0.7, 4 and 6 mN, respectively). Red arrows indicate these specific events. The yellow star in (*C*) marks an artefact created by the 0.4 mN monofilament passing above the OCT light beam. The right panel illustrates the average skin displacement across the conditions (\*\**P* < 0.01, \**P* < 0.05). [Colour figure can be viewed at wileyonlinelibrary.com]

to be tightly linked anatomically to the follicle, similar to the relationship between C-LTMRs and awl/auchene and zigzag hairs in mice (Li et al., 2011). Given that skin deformation induced by hair deflection is minimal, and less than that induced by low force Von Frey mono-filaments, an anatomical connection to the hair follicle appears probable. In either case, the strong expression of Piezo2 in CTs/human C-LTMRs documented by single cell transcriptomics from human DRG (Yu et al., 2024) will facilitate this mechanosensitivity. A further possibility is that chemical transmission could contribute to afferent firing during hair deflection. It has been shown that outer root sheath cells in human hair follicles can release serotonin and histamine and that these transmitters can activate co-cultured with murine sensory neurons, as detected by calcium imaging. Although sensory neuron subtypes were not defined, and whether the same would occur in human neurons is unknown, it does suggest the viability of a chemical signalling mechanism in addition to a mechanism mediated by mechanosensitive ion channels such as Piezo2 (Agramunt et al., 2023).

There are a number of limitations regarding the methodology and interpretation of our study. Hair deflection and pulling were mechanically delivered by hand and therefore were not a controlled stimulus. This may have introduced variability in mechanical force. Because stimuli were delivered by hand rather than in a more controlled manner using a robot, the delivery had reduced timing precision, therefore introducing variation in duration and speed. Finally, it has recently been shown that TH+ C-LTMRs in mice show directional sensitivity and fire maximally to stimuli delivered against the natural orientation of the hair (Semizoglou et al., 2025). Our data do not explicitly indicate that all CTs innervate hair follicles, but rather, that CTs can innervate hair follicles. All 15 CT reported here respond to hair deflection without significantly inducing perifollicular skin displacement, suggesting an association with the hair follicle. However, it would not be appropriate to suggest that this is either characteristic of 100% of all human CTs or comprise a CT subpopulation without further research based on a larger sample size.

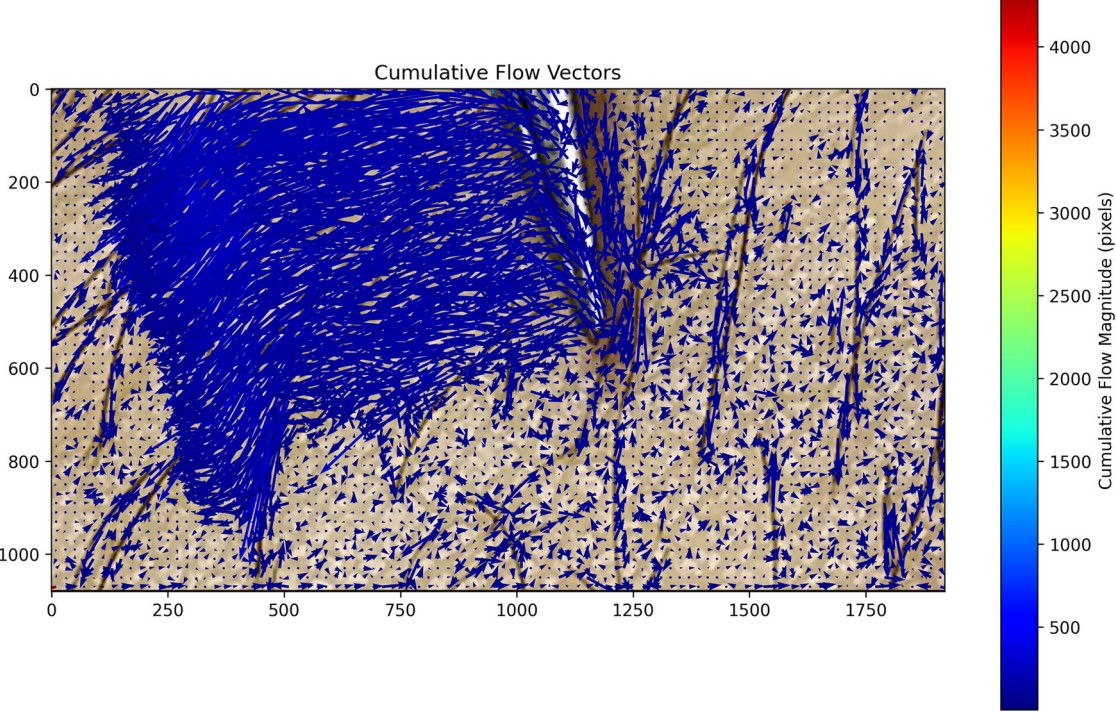

**Figure 8. Cumulative flow vector analysis during targeted hair deflection**
Vector field visualization showing cumulative displacement during controlled hair deflection using forceps. Blue arrows indicate direction and relative magnitude of total displacement, computed using the Farnebäck optical flow algorithm (1920 × 1080 pixels, 60 frames s$^{-1}$). The prominent flow pattern in the central region represents the path of forceps movement and associated hair displacement. Notably, the surrounding skin tissue shows minimal flow magnitudes comparable to background levels, indicating that mechanical hair deflection does not induce substantial skin deformation around the stimulated hair. The colour bar (right) indicates cumulative flow magnitude in pixels. [Colour figure can be viewed at wileyonlinelibrary.com]

To further understand the involvement of hairs within the receptive field of CTs responses should be compared before and after hair removal. Recording CT response to hair deflection of different velocities would further improve our understanding of how hair movement contributes to CT responses. A more extensive investigation into hair-skin mechanics would allow for greater precision when characterizing the minute hair deflection induced skin deformations.

In conclusion, human CTs respond to hair deflection and exhibit ongoing activity following mechanical hair removal, lasting ~7.1 s. Overall, the findings suggest an anatomical coupling between the CT and hair follicle. Future research aiming to label CTs/C-LTMRs in human hairy skin should focus on detecting putative fibres in close anatomical proximity to hair follicles.

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

## Additional information

### Data availability statement

Datasets are available from the corresponding author upon reasonable request. Data transfer agreements and associated costs may be required.

### Competing interests

The authors declare that they have no competing interests.

## Author contributions

A.M., S.N. and H.O. conceived and designed the microneurography experiments. O.B., A.M., S.N., AMak and W.M, conducted the microneurography experiments. W.M. and J.N. analysed the microneurography data. The related microneurography plots, methods and results were written by W.M., which were edited by A.M., S.N., H.O., J.N. and F.M. The OCT experiments were conducted by B.D., S.M, A.F. and L.P. Both B.D. and S.M. equally contributed to the data analysis and writing of the results. P.R. and M.R. contributed to conducting, analysing and writing the results for the optical flow computation experiment. W.M., J.N, O.B., P.H., M.R., B.D., S.M., A.F., F.M., H.O., S.N. and A.M contributed to writing of the manuscript. All authors participated in the critical reading of the manuscript and gave their consent for the final draft. All authors have approved the final version of the manuscript submitted for publication and agreed to be accountable for all aspects of the work. All persons designated as authors qualify for authorship, and all those who qualify for authorship are listed.

## Funding

This work was supported by ALF Grant Region Östergötland, Swedish Research Council Project Grant (2021-0 3054), The Swedish Society for Medical Research (SSMF), Swedish Research Council (2020-0 1085, 2024-00381) and the Pain Relief Foundation (Liverpool).

## Acknowledgements

We thank Bengt Ragnemalm and Ilona Szczot for technical assistance with stimulus delivery for OCT imaging.

## Keywords

C-tactile afferent, hair movement, hair follicle, microneurography

## Supporting information

Additional supporting information can be found online in the Supporting Information section at the end of the HTML view of the article. Supporting information files available:

**Peer Review History**
**Supporting Information**
**Supporting Information**

