## [Peer Review History · The Journal of Physiology]

Robust Coupling Between the C-Tactile Afferent and the Hair Follicle in Humans

Warren Moore, Johan Nikesjö, Otmane Bouchatta, Adarsh D Makdani, Pierre Hakizimana, Mikael Rousson, Basil Duvernoy, Sarah McIntyre, Laura J Pehkonen, Anders Fridberger, Francis McGlone, Hakan Olausson, Saad S Nagi, and Andrew Marshall

DOI: 10.1113/JP287706

Corresponding author(s): Andrew Marshall (andrew.marshall@liverpool.ac.uk)

The following individual(s) involved in review of this submission have agreed to reveal their identity: Ingvars Birznieks (Referee #1)

Review Timeline:

Submission Date:	24-Oct-2024
Editorial Decision:	20-Nov-2024
Revision Received:	16-Apr-2025
Editorial Decision:	08-May-2025
Revision Received:	05-Jun-2025
Accepted:	13-Jun-2025

Senior Editor: Nathan Schoppa

Reviewing Editor: Vaughan Macefield

Transaction Report:

Dear Dr Marshall,

Re: JP-RP-2024-287706 "An Investigation into the Putative Coupling Between the C-Tactile Afferent and Hair Follicles as Revealed with Microneurography" by Warren Moore, Johan Nikesjö, Otmane Bouchatta, Adarsh D Makdani, Pierre Hakizimana, Mikael Rousson, Francis McGlone, Hakan Olausson, Saad S Nagi, and Andrew Marshall

Thank you for submitting your manuscript to The Journal of Physiology. It has been assessed by a Reviewing Editor and by 2 expert referees and we are pleased to tell you that it is potentially acceptable for publication following satisfactory major revision.

REVISION CHECKLIST:

We look forward to receiving your revised submission.

Yours sincerely,

Nathan Schoppa
Senior Editor
The Journal of Physiology

REQUIRED ITEMS

- Author photo and profile. First or joint first authors are asked to provide a short biography (no more than 100 words for one author or 150 words in total for joint first authors) and a portrait photograph. These should be uploaded and clearly labelled together in a Word document with the revised version of the manuscript. See Information for Authors for further details.

- The contact information for the person responsible for 'Research Governance' at your institution needs to be provided. This includes their name and an institutional email address. Please ensure the contact is not an author on this paper and provide an alternate contact if necessary, or confirm in the submission form that the author whose email was provided has sole responsibility for research governance. This is the person who is responsible for regulations, principles and standards of good practice in research carried out at the institution, for instance the ethical treatment of animals, the keeping of proper experimental records or the reporting of results.

- You must start the Methods section with a paragraph headed Ethical Approval. If experiments were conducted on humans, confirmation that informed consent was obtained, preferably in writing, that the studies conformed to the standards set by the latest revision of the Declaration of Helsinki and that the procedures were approved by a properly constituted ethics committee, which should be named, must be included in the article file. If the research study was registered (clause 35 of the Declaration of Helsinki), the registration database should be indicated, otherwise the lack of registration should be noted as an exception (e.g. The study conformed to the standards set by the Declaration of Helsinki, except for registration in a database). For further information see: <https://physoc.onlinelibrary.wiley.com/hub/human-experiments>.

- Your manuscript must include a complete Additional Information section, including competing interests; funding; author contributions and acknowledgements.

- The Journal of Physiology funds authors of provisionally accepted papers to use the premium BioRender site to create high resolution schematic figures. Follow this link and enter your details and the manuscript number to create and download figures. Upload these as the figure files for your revised submission. If you choose not to take up this offer, we require figures to be of similar quality and resolution. If you are opting out of this service to authors, state this in the Comments section on the Detailed Information page of the submission form. The link provided should only be used for the purposes of this submission. Authors will be charged for figures created on this premium BioRender account if they are not related to this manuscript submission.

- Please upload separate high-quality figure files via the submission form.

- Please ensure that any tables are editable and in Word format, and wherever possible, embedded in the article file itself.

- Please ensure that the Article File you upload is a Word file.

EDITOR COMMENTS

Reviewing Editor:

Ethics Concerns:

Please indicate that the studies conformed with the Declaration of Helsinki, with except for registration in a database.

Comments to the Author:

Thank you for submitting your manuscript to The Journal of Physiology. I have now received comments from two independent reviewers, both experts in microneurography. Both reviewers see merit in your study yet both have reservations, especially Reviewer 2. The primary issue is whether you can be certain that these afferents are responding to hair deflection, as opposed to skin displacement; as Reviewer 2 suggests, chemical removal of the hair follicles may allow this to be addressed though I recognise that this may be difficult to achieve during the actual experiment. Either way, you need to convince the reviewers and myself that this manuscript offers something that is novel and advances our understanding of the neurophysiology of CT afferents. I invite you to revise the manuscript extensively and submit point-by-point responses to the reviewers' comments. I look forward to receiving your revised manuscript in due course.

Senior Editor:

Comments to the Author:

Thank you for submitting your manuscript to Journal of Physiology. The work has been evaluated by two expert reviewers and reviewing editor, who believe that the work could be potentially impactful but not in its present form. Several important points have been raised that will require at a minimum additional analyses and considerably more information about methodological details and controls to convince reviewers of the novelty of the results. We invite you to submit a manuscript that addresses all points that were raised, after which we would re-evaluate your study.

REFEREE COMMENTS

Referee #1:

This study reports on 12 human CT (C-LTMR) afferent responses to selective hair deflection in a similar fashion as to slow brush movements and exhibiting a prolonged response to mechanical hair removal. This is quite a novel finding, as previously it was believed that CT units are free nerve endings in the hairy skin with no association with hair follicles.

Nevertheless, in the overall context, this finding might be logically expected. In animals, the association between C-LTMR afferents and hair follicles is well established; however, there is no evidence that these receptors are the same type (TH and MrgprB4 expression seems to differ). It has also been intriguing why, in humans, CT afferents are found only in the hairy skin and not in the glabrous skin. Therefore, these new findings in the current study make a lot of sense, but at the same time, it must be discussed and explained why this has not been observed previously over more than two decades of research. The manuscript doesn't attempt to engage in such discussion.

The manuscript doesn't report whether these are a specific subset of CT afferents and in what fraction of all human C-LTMR afferents an association with hair follicles could be detected if specifically tested for.

It has to be confirmed whether each afferent is always innervating a single hair follicle.

How were mechanical thresholds measured? Did stimulation of the skin around the hair follicle evoke a response in these

afferents? What's the topography of sensitivity profiles? It would be useful to discuss how it matches with receptive field characteristics reported in the literature previously. Did C-LTMR afferents respond to mechanical stimuli after hair removal and after-discharge settled?

Please include statement whether the study conformed to the Declaration of Helsinki and whether a written consent was obtained from the participants.

The Microneurography and Neural Data Analyses sections are repetitious. It is not clear why R is mentioned in the Microneurography section, as spike sorting was not done in R.

In the Unit Classification section (P9), the statement "below 4mN" contradicts the results mentioning 5mN.

Why was the response defined as a minimum of 2 spikes (P10)? One spike in a population of afferents can provide meaningful temporal information.

Why were responses investigated in one direction only? Did afferents respond when the hair was moved back to its original position?

How was the mean instantaneous frequency measured? From the first to the last spike or over a predefined stimulation time period?

What were the HFA mechanical thresholds?

In Figure 2, the timing of the stimulus is not indicated.

In Figure 3, there are 4 pairs of data, but comparison stats are given only for one of them.

On P14, the second last sentence - reporting of stats should be consistent within the same sentence.

The Discussion, third paragraph, is a repetition of the Introduction section.

On P18, the statement that "data does not explicitly indicate that all CTs innervate hair follicles, but rather, CTs can innervate hair follicles" is very vague. More than that should be said. Does it mean that some CTs just happen to be in the vicinity of hair follicles? It is not the same as saying they are hair follicle associated afferents. If there is nothing more to say, maybe the title of this paper should be "C-Tactile afferents respond to hair deflection"?

Or do you believe this a new different type of human C-LTMR afferents than reported elsewhere?

As mentioned above, at least some rough idea of the fraction of CT afferents that would exhibit a response to hair deflection should be given.

Referee #2:

The paper from Moore et al. presents recordings from C-tactile (CT) afferents to hair deflection and relates this functionally to the hairs. Papers on CT afferents in humans always provide highly valuable data and I appreciate that the authors have distinguished between CT and C-low threshold mechanoreceptor (C-LTMR) as terms relating to humans and other animals, respectively. It helps make sense of the evidence and differences between species. In the present study, it is also very elegant that the authors have combined physiological recordings with imaging of the hair movements. On the surface, the work is interesting and novel, yet there are a number of considerations that need to be taken into account.

Overall, there is no definition of the stroking velocity, which acts as a control comparison condition, but this is an important factor for CTs that change their firing according to stroking speed. The authors do not even give a range or approximation of what 'slow stroking' actually meant in their study. This makes it difficult to compare to previous work and also challenging to relate to the hair deflection results.

The paper claims that slow brushing over a CT receptive field gives a similar response to hair pulling. This is going too far, as to say that these things are equivalent means that you controlled both duration of stimulation and the velocity (for stroking and hair movement), which does not seem to have been done. As you are basing your results off firing frequency, it is important to know the duration of stimulation, which I guess you worked out via the time between the first and last spike. There are a number of points to look at here, but the main ones are that your overall firing frequencies seem very low (is it a problem with the timing calculation for the frequency?) and that you should actually have the timing data for both stroking and hair pull that can be applied to your analysis.

Following on from this, the authors say in the discussion that slow, gentle stroking is the previously documented preferential stimulation, but they now imply that hair bending is just as effective a stimulus. I would not exactly claim this, as the speed of hair bending was not investigated (e.g. to be equivalent to stroking velocity in speed/duration). A number of previous CT papers have also shown that indentation gives equivalently high firing frequencies in CTs (e.g. Vallbo et al, 1999, J Neurophysiol). It therefore seems that any low force mechanical displacement in the skin activates CTs, but in essence, they are mechanoreceptors. The question is more, how sensitive are CTs (i.e. minimum skin displacement)? Is it just that the hair displaces the skin and CTs are very sensitive? This was already postulated by Iggo & Kornhuber (1977, J Physiol).

Concerning the response firing rates, the data presented do not seem to be of typical physiological response rates for CTs to one of its preferred stimuli. All other CT papers show much higher firing rates for all other touch stimuli. If it really is the case that you find much lower firing frequencies, to hair deflection or stroking, this must be discussed.

Specific points in the paper:

Abstract and key points

It says, "C-tactile afferents... have a functionally relevant anatomical relationship with hair follicles". This may be taking your message too far. You show that CTs can be activated via hair deflection and hair plucking, but this all relates more specifically to changes in skin displacement. You cannot be sure of whether it is truly hairs or just the skin, you did not look at the anatomical relationship and you did not chemically remove the hair. Although the hair plucking hints that CTs are coupled to hairs, there is still a big skin displacement, thus it seems like this is still an open question and it could just be a correlation between skin displacement and firing.

Introduction

In the introduction, there are a few papers that need to be included to introduce exactly what we know about the responses of CTs/C-LTMRs to hair deflection. In cats, Iggo (1960, J Physiol) showed that some C-LTMRs are sensitive to hair movement, but not others. In a follow-up study, Iggo & Kornhuber (1977, J Physiol) said that transmission of the deformation of the skin by hair displacement increased C-LTMR firing in cats (summarized). They even quantified that very small hair movements in the range of 10-20 μm could excite C-LTMRs. They also brought up the interesting point that afterdischarge could be directly related to hair displacement (and replacement), although hairs were not a sufficient factor to see this afterdischarge that still occurred on hair removal. Therefore, it is difficult to conclude the exact structural and functional relationship between CTs/C-LTMRs, as they are clearly co-related, but hairs are not sufficient for C-LTMR responses. Concerning human literature, three papers appear to mention CT responses to hair movement. Olausson et al (2010, Neurosci Biobeh Rev) stated that qualitative observations indicate that most CTs are not specifically sensitive to hair movements. Nordin (1990, J Physiol) found that, on the face, a single unit with scalp hairs arising from its receptive field responded vigorously to hair movements, but there was little or no response to sustained displacement of hairs; however, this was just one CT and scalp hairs can be rather thick. Further, Ackerley et al (2014, J Neurosci) stated that CTs did not respond to air puffs, which activated hair afferents. Therefore, your work needs to be put into perspective in the introduction (and discussion) in relation to these previous findings that in animals, C-LTMRs sometimes elicited a response from hair deflection and in humans, there was little evidence for a response from CTs to hair movement.

In the middle of the introduction, when you write about hairs being present on the glabrous skin of the rodent paw (Walcher et al, 2018), it seems important to add that this is the hindpaw. The reference explicitly states that the hairs in glabrous mouse skin are very few and that they are only found in the hindpaw. Therefore, it is imprecise to make a point of mice having glabrous hair afferents when there really are very few and it is only on their hind feet.

Methods - participants

Did the participants sign an informed consent form? Did it conform to national and international regulations for human experiments?

Methods - hair deflection

Can you give more precise information about how many repetitions of stimuli were given?

Methods - tracking hair deflection

This part is very interesting, yet there is little information on how this was done (e.g. camera, spatial and temporal resolution of videos, how an individual hair was actually tracked, software for tracking/analyzing). Further, the results of the tracking are not at all used and this would be extremely interesting to correlate the speed (and/or duration) of the bending to the firing of the units.

Methods and results

You write about 'slow brush stroking', but this is vague. It is difficult to interpret the data when the actual speed, or at least an estimation of it, is not provided. This impacts on the results for firing frequency, which should change depending on stroking speed for all mechanoreceptors. Typically, any stroking from 0.3 to 30 cm/s would show mean firing rates of at least 20 spikes/s for CTs, whereas hairs would be from about 10 to 150 spikes/s (Löken et al, 2009; Ackerley et al, 2014). You state that slow stroking gave median firing rates of just 14 spikes/s for CTs and 37 spikes/s for HFAs, which is atypical for both. Did you track the stroking via video, like you tracked the hair movements or have some way of marking the speed of the stroking? If not, this is a variable that is not controlled and could have implications for the interpretation.

Results

You mention in the results that you also tested air puff responses of CTs (which is not in the methods). As said above, CTs have not typically been shown to respond to air puffs, therefore this should be better explained. One thing that would again be useful to have is the timing, especially with regards to the analysis of firing frequency. It seems that if you are unsure of stimulus timings, peak frequency could be a good option to focus on and/or the response in the first second only (as long as the stimulus was longer than a second).

Results - hair plucking

You say that you tested 3 HFAs to hair plucking, but do not give any further information. Could you also add the same data as per the CTs, i.e. the instantaneous frequency data, and ideally add an example to Fig. 4?

Figures - overall

The figures would benefit from better scale bars. For example, the horizontal scales for spike trains could have time markers or at least the scale bar at the beginning of the spike firing, as would be more typical. Also, the vertical scales often only consist of one or two points, with no line indicating the axis. A more indicative scale would be much more informative, especially considering the question over the firing rates.

Figures - Fig. 3

Why is the vertical scale in Fig. 3 log₁₀? This seems quite difficult to interpret, especially in comparison to the literature. I guess it is for the comparison between CTs and HFAs, but it looks like the results would be even clearer on a linear scale. Again, it seems strange that CTs would produce such low mean firing frequencies to stroking.

Figures - Fig. 5

Could you add the actual CT recording to Fig. 5?

Discussion

In the part about monofilaments and CT receptive fields in relation to hairs, you need to develop these points further. You say the 'mechanical detection threshold of CTs using monofilaments', but this must not be confused with perception, so I suggest to use a better term, which could be the 'force activation threshold'. As the hairs appear to act as a support for mechanoreceptors in hairy skin, it is no surprise that hair deflection could activate any mechanoreceptor, thus this point about monofilaments would surely be the same for measuring thresholds for all mechanoreceptors. Also, CT force activation thresholds are so low (e.g. often well less than 5 mN), would moving a hair at this force really have an effect on CT firing? Further, what is your point about the receptive field in relation to the Wessberg et al (2003) paper, why would your results explain the maps of 1-9 hot spots? Again, I am not convinced that the gentle bending of a hair only could really generate the responses seen, although it could contribute. The receptive fields detailed in the Wessberg paper would be expected from mechanoreceptors and C-fibers; would you predict something different on hair removal? This is a point that could be followed up on.

In the paragraph starting, "It may be suggested that hair deflection...", I am not sure what you point is about skin deformation and other hairs touching is, and how HFAs have the same caveat. Even if you move a single hair, it will likely cause the underlying skin to move.

Overall, the discussion does raise some interesting points, but it could be shorted in some parts to include a number of important things have not been considered, e.g.:

- Why hair deflection would be such a pertinent stimulus for CTs
- Why your results are different to literature that indicates some sensitivity of C-LTMRs to hair deflection and little sensitivity in CTs. Also, it is not clear if the sample you present is true for all CTs, i.e. whether you only document CTs that have a positive hair deflection response or whether you see CTs that are unresponsive.
- Why bending a hair gives a much stronger response in CTs than the actual HFA.
- That there are inherent difference between the thickness of hairs (fur) between humans and other animals, which could produce different results.
- There is also the point that the association between CTs and hair could be true for all mechanoreceptors in hairy skin (cf. Li et al, 2011), where the hairs provide an anchoring matrix structure, but taking it further, it does not mean that mechanoreceptors require hairs to function.
- A discussion of further experiments, e.g. speed of hair bending and if this relates to the typical CT inverted-U curve, chemical hair removal.

Minor comments

- Firing frequency is stated in Hz, but it is typically in spikes or impulses per second for physiology.
 - In the results, you write a few times things like 'mean frequency: Mdn = ...'. Do you mean the group median of the mean frequency per unit? This does not seem to make much sense. You could simply just present the group means.
 - Please add page and/or numbers to your paper to help follow with the review.
-

END OF COMMENTS

Dear Editor,

Firstly, we would like to thank the reviewers for taking the time to provide in-depth and nuanced feedback. We found the feedback extremely useful, and we feel that the careful revisions in response have significantly enhanced the clarity and quality of the manuscript. Guided by the reviewer's excellent suggestions, we have made substantial modifications and additions to the manuscript. To address the concern of whether CT nerve responses are a result of hair movement induced skin displacement two experiments (and an additional four co-authors) were included, optical flow computation and optical coherence tomography. Optical flow computation refers to the process of interpreting motion patterns of objects within a video recording via tracking of individual pixels across successive frames. This method allowed us to estimate the velocity at which individual hairs were deflected using forceps ($M = 0.8\text{cm/s}$). In addition, this method tracked the movement of both surrounding hairs and skin displacement, which indicated minimal peri-follicular skin movement during hair deflection (See supplementary video S1 and S2). Optical coherence tomography - is a non-invasive imaging technique that uses near-infrared light to capture high-resolution, cross-sectional images of the skin. The purpose of this experiment was to quantify the amount of surface skin displacement during monofilament indentations, hair deflection, and when the skin is at rest. The results indicated that the bending of a hair induced less surface skin deformation than a 0.4mN monofilament indentation. In combination, we feel that the additional evidence both addresses the reviewers' concerns and strengthens the argument for an anatomical association between the CT and hair follicle in humans. Our point-by-point responses to the editorial and reviewer comments are set out below. Bold red text indicates text that has been added to the manuscript.

Reviewing Editor:

Ethics Concerns:

Please indicate that the studies conformed with the Declaration of Helsinki, with except for registration in a database.

This has been added: "This study conformed to the Declaration of Helsinki and written consent was obtained from each participant" (page 5).

REFEREE COMMENTS

Referee #1:

This study reports on 15 human CT (C-LTMR) afferent responses to selective hair deflection in a similar fashion as to slow brush movements and exhibiting a prolonged response to mechanical hair removal. This is quite a novel finding, as previously it was believed that CT units are free nerve endings in the hairy skin with no association with hair follicles.

Nevertheless, in the overall context, this finding might be logically expected. In animals, the association between C-LTMR afferents and hair follicles is well established; however, there is no evidence that these receptors are the same type (TH and MrgprB4 expression seems to differ). It has also been intriguing why, in humans, CT afferents are found only in the hairy skin and not in the glabrous skin. Therefore, these new findings in the current study make a lot of sense, but at the same time, it must be discussed and explained why this has not been observed previously over more than two decades of research. The manuscript doesn't attempt to engage in such discussion.

Thank you for these comments. To enrich the discussion section we have added: **“Data on the responsiveness of CTs to hair movement in humans are sparse, and we can only speculate on why our results indicate a greater CT sensitivity to hair deflection in comparison to previous literature. We undertook a rigorous systematic exploration of 15 consecutively recorded CTs where a single hair was carefully deflected under magnification, and perhaps previous studies have not focused on hair deflection responses in detail (page 22).**

The manuscript doesn't report whether these are a specific subset of CT afferents and in what fraction of all human C-LTMR afferents an association with hair follicles could be detected if specifically tested for.

Thank you for this comment. It is possible be more than one population of CTs. However, we tested 15 **consecutive** CTs where the fibres were held for long enough to allow testing. Clearly we cannot comment about fibres that were lost before we could test responsiveness to hair deflection. In the discussion previously stated “Our data does not explicitly indicate that all CTs innervate hair follicles, but rather, CTs can innervate hair follicles.” This has now been amended to **“Our data does not explicitly indicate that all CTs innervate hair follicles, but rather, CTs can innervate hair follicles. All 15 CT reported here respond to hair deflection without significantly inducing perifollicular skin displacement, suggesting an association with the hair follicle. However, it would not be appropriate to suggest that this is either characteristic of 100% of all human CTs or are a CT subpopulation without further research based on a larger sample size”** (page 24).

It has to be confirmed whether each afferent is always innervating a single hair follicle.

Thank you for this comment. There were instances when we were able to separately test deflection of more than one individual hair within a ‘receptive field’. We were able to show that CTs do fire in response to deflection of multiple single hairs.

A figure has been added showing the raw neural response to hair deflection of two individual hairs innervating the same receptive field of a single CT unit, indicating that a CT can respond to movement of more than one individual hair. Furthermore, we have added to the text in the methods: **“When possible, the firing response to deflection of more than one individual hair was studied”** page(8). And the following to the results: **“Hair deflection response of a single CT innervated by two individual hairs was recorded. Both hair one (mean instantaneous frequency = 7 and peak instantaneous frequency = 13) and hair two (mean instantaneous frequency = 23**

and peak instantaneous frequency = 40) displayed hair deflection sensitivity (Figure 3)” (page 14-15).

How were mechanical thresholds measured?

“Monofilaments (0.08-3000mN) were used to determine adaptation type and mechanical threshold” (page 6). Units were considered to respond if a response was observed on at least 50% of occasions to monofilament indentation.

Did stimulation of the skin around the hair follicle evoke a response in these afferents?

Other than for measuring mechanical threshold, where the skin within the receptive field was stimulated at a specific hotspot, we did not attempt to mechanically stimulate the skin specifically around the hair follicle itself.

What's the topography of sensitivity profiles? It would be useful to discuss how it matches with receptive field characteristics reported in the literature previously.

Thank you for this comment. We did not examine this in the current study. This has been published previously by Wessberg et al (2003) in a technical tour-du-force examining 9 CT afferents with a robotic stimulator. We did mention this in the discussion and speculated the hotspots of sensitivity for CT activation may be related to the location of hair follicles “It could also explain their receptive field properties with several hotspots often being noted within the receptive field (Wessberg et al 2003).” For clarity this has been amended to “Given that CTs can innervate more than one follicle within a receptive field, it could also explain the receptive field properties of CTs which typically exhibit several hotspots of mechanical sensitivity, which could potentially reflect the location of hairs or their follicles, within the receptive field (Wessberg et al 2003)” (page 22).

Did C-LTMR afferents respond to mechanical stimuli after hair removal and after-discharge settled?

Thank you for this question. Since CTs respond to the separate deflection of more than one individual hair within the receptive field this is very difficult to test with any certainty as the mechanical stimulus, whether this be a vonFrey monofilament or soft brush, could cause movement of other hairs within the receptive field. It is neither practical nor possible to ensure that all hairs within the receptive field have been removed. A statement to this effect has been added to the results

“CTs remained mechanically responsive after removal of individual hairs” (page 16).

and to the discussion

“CTs remained sensitive to soft brushing and/or vonFrey monofilament indentation following the removal of individual hairs. However, it is not possible to know whether these were responses to direct skin stimulation or due to movement of other hairs within the receptive

field. We did not assess if removal of hairs changed the receptive field properties, e.g., altered the number or sensitivity of the hotspots described by Wessberg et al., (2003)” (page 22).

Please include statement whether the study conformed to the Declaration of Helsinki and whether a written consent was obtained from the participants.

Added the following statement to the methods section:

“This study conformed to the Declaration of Helsinki and written consent was obtained from each participant” (page 5).

The Microneurography and Neural Data Analyses sections are repetitious. It is not clear why R is mentioned in the Microneurography section, as spike sorting was not done in R.

The reference to R has been removed from the microneurography section. Repetitions within the neural data analysis section have been removed and spike sorting details added. The microneurography section has been amended to:

“Captured neural data were analysed using LabChart (v8.1.28, ADInstruments, New Zealand), where threshold crossing was used to distinguish action potentials from noise with spike morphology confirmed by template matching (spike height, width, and shape) to action potential shape during soft brushing trials. For each stimulus, spike count, mean instantaneous frequency, (calculated between the first and last spike) and peak instantaneous frequency were calculated. Recordings were discarded if multiple units were present (e.g., non-physiological spike intervals/firing rates) or if spike amplitudes were not distinct from the noise, preventing secure action potential identification. The neural response was evoked (or modulated) only when the specific area of skin (the receptive field) was stimulated” (page 6-7).

In the Unit Classification section (P9), the statement "below 4mN" contradicts the results mentioning 5mN.

Thank you for spotting this. During recording units were classified as LTMR if they were soft brush responsive and during analysis we used a 4mN or below cut-off for classification. This has been amended throughout the manuscript to “4mN or below”.

Why was the response defined as a minimum of 2 spikes (P10)? One spike in a population of afferents can provide meaningful temporal information.

A minimum of 2 spikes was chosen to provide an extra layer of confidence that the response seen was an evoked one and also because at least two spikes are needed to calculate an instantaneous firing frequency.

Why were responses investigated in one direction only? Did afferents respond when the hair was moved back to its original position?

Hair deflection experiments were technically demanding, and we were fastidious in the way we deflected the hair to ensure there was no contact with the skin or movement of adjacent hairs (that may touch the skin and stimulate the fibre being recorded). This meant that we focussed on

hair deflection in a single direction, against the direction of the hair (note we cannot exclude some contamination with movement back to the original position).

How was the mean instantaneous frequency measured? From the first to the last spike or over a predefined stimulation time period?

This was calculated between first and last spike. The methods section now includes the following statement: **“For each stimulus, spike count, mean instantaneous frequency, (calculated between the first and last spike) and peak instantaneous frequency were calculated”** (page 6-7).

What were the HFA mechanical thresholds?

As in Vallbo et al (1995), we did not measure mechanical thresholds for HFA. HFAs responded to hair deflection (air puff, manual hair deflection) and since you can catch a hair with the vonFrey monofilament and produce spiking, thresholds are unreliable (Vallbo et al, 1995). In fact, because we showed that CT afferents responded to hair deflection, we made the argument that testing of their mechanical thresholds is problematic for the same reason. For clarity we have added the following text to Methods: **“Among A β RA-LTMRs, units that responded to hair deflection were identified as HFAs as per criteria used by Vallbo et al (1995)”** (page 7).

In Figure 2, the timing of the stimulus is not indicated.

Stimulus timing bars have been added (blue bars) to figure 2 now more clearly indicating the onset, offset and duration of the stimulus.

In Figure 3, there are 4 pairs of data, but comparison stats are given only for one of them.

Comparison statistics have been added to compare hair deflection and brushing for both CTs and HFAs based on mean frequency and peak frequency responses to hair movements. Therefore, four sets of comparisons have been added as recommended and the caption now reads: **“(a) CT mean instantaneous frequency and peak instantaneous frequency response to brushing and hair deflection. The response to brushing and hair deflection was statistically indistinguishable (Mann-Whitney test: mean frequency, $p = .057$; peak frequency: $p = .104$). (b) HFA mean instantaneous frequency and peak instantaneous frequency response to brushing and hair deflection. The response to brushing and hair deflection was statistically indistinguishable (Mann-Whitney test: mean frequency, $p = .250$; peak frequency: $p = .250$ ”** (page 16).

On P14, the second last sentence - reporting of stats should be consistent within the same sentence.

This has been amended to report statistics in a consistent format.

The Discussion, third paragraph, is a repetition of the Introduction section.

This has now been amended and the discussion within this paragraph is in the context of the current findings.

On P18, the statement that "data does not explicitly indicate that all CTs innervate hair follicles, but rather, CTs can innervate hair follicles" is very vague. More than that should be said. Does it mean that some CTs just happen to be in the vicinity of hair follicles? It is not the same as saying they are hair follicle associated afferents. If there is nothing more to say, maybe the title of this paper should be "C-Tactile afferents respond to hair deflection"?

This has now been amended to: **“Our data does not explicitly indicate that all CTs innervate hair follicles, but rather, CTs can innervate hair follicles. All 15 CT reported here respond to hair deflection without significantly inducing perifollicular skin displacement, suggesting an association with the hair follicle. However, it would not be appropriate to suggest that this is either characteristic of 100% of all human CTs or are a CT subpopulation without further research based on a larger sample size”** (page 25).

Optical Coherence Tomography showed that skin movement during gentle hair deflection was minimal and not significantly greater than during no stimulation. This suggests that CTs were particularly sensitive to hair deflection.

The title of the manuscript has been altered to **“Robust Coupling Between the C-Tactile Afferent and the Hair Follicle in Humans”** (page 1).

Or do you believe this a new different type of human C-LTMR afferents than reported elsewhere? **We found a positive hair deflection response in 15 consecutive CTs with a visible hair in the receptive field, and we feel it is unlikely that a new type of CT would be consecutively recorded so frequently.**

As mentioned above, at least some rough idea of the fraction of CT afferents that would exhibit a response to hair deflection should be given.

In our sample, all 15 CTs responded to hair deflection, however this is not a large enough sample size to speculate on a percentage estimate.

Referee #2:

The paper from Moore et al. presents recordings from C-tactile (CT) afferents to hair deflection and relates this functionally to the hairs. Papers on CT afferents in humans always provide highly valuable data and I appreciate that the authors have distinguished between CT and C-low threshold mechanoreceptor (C-LTMR) as terms relating to humans and other animals, respectively. It helps make sense of the evidence and differences between species. In the present study, it is also very elegant that the authors have combined physiological recordings with imaging of the hair movements. On the surface, the work is interesting and novel, yet there are a number of considerations that need to be taken into account.

Overall, there is no definition of the stroking velocity, which acts as a control comparison condition, but this is an important factor for CTs that change their firing according to stroking speed. The authors do not even give a range or approximation of what 'slow stroking' actually meant in their study. This makes it difficult to compare to previous work and also challenging to relate to the hair deflection results.

Added statement indicating that brushing was “delivered manually by a trained experimenter with a soft brush at a velocity of approximately 3cm/s” (page 7).

The paper claims that slow brushing over a CT receptive field gives a similar response to hair pulling. This is going too far, as to say that these things are equivalent means that you controlled both duration of stimulation and the velocity (for stroking and hair movement), which does not seem to have been done. As you are basing your results off firing frequency, it is important to know the duration of stimulation, which I guess you worked out via the time between the first and last spike. There are a number of points to look at here, but the main ones are that your overall firing frequencies seem very low (is it a problem with the timing calculation for the frequency?) and that you should actually have the timing data for both stroking and hair pull that can be applied to your analysis.

The low instantaneous frequency calculated was due to a calculation issue, where instantaneous frequency was calculated from the push button onset/offset markers rather than between the first and last neural spike. After calculating the mean instantaneous frequency between the first and last spike during the stimulation period – the instantaneous frequency is now closer to the expected value of 21 spikes/second. In addition, with the same adjustment HFA brushing instantaneous frequency was recalculated as 31 spikes/second. The appropriate text and figures have been amended with the updated values. While we accept that the mean instantaneous frequency is still lower than previously published observations, the peak instantaneous frequencies (CT: M = 44 spikes/second; HFA: M = 70 spikes/second) are within the expected range (Löken et al., 2009, Nature Neuroscience). Therefore, focusing on the peak instantaneous frequency overcomes the issue of approximate stimulus timings, and indicates that all 15 CTs presented show a soft brush response that is within the typical range.

Following on from this, the authors say in the discussion that slow, gentle stroking is the previously documented preferential stimulation, but they now imply that hair bending is just as effective a stimulus. I would not exactly claim this, as the speed of hair bending was not investigated (e.g. to be equivalent to stroking velocity in speed/duration). A number of previous CT papers have also shown that indentation gives equivalently high firing frequencies in CTs (e.g. Vallbo et al, 1999, J Neurophysiol). It therefore seems that any low force mechanical displacement in the skin activates CTs, but in essence, they are mechanoreceptors. The question is more, how sensitive are CTs (i.e. minimum skin displacement)? Is it just that the hair displaces the skin and CTs are very sensitive? This was already postulated by Iggo & Kornhuber (1977, J Physiol).

Thank you for this important comment. We have expanded upon potential reasons how hair deflection may induce firing of CTs in the discussion including a comparison with Field afferents which (at least in rodents) have an anatomical association with hair follicles yet do not respond to

hair deflection. The OCT data are highly relevant in this respect and the following has been added to directly address this important issue:

“Only two (of fifteen) CTs had a mechanical threshold as low as 0.4mN, which our optical coherence tomography experiment confirmed produces 17µm skin displacement. Given that hair deflection induced skin displacement (8µm) was significantly less than this (8µm - approximately half the displacement for that of a 0.4mN monofilament), it is reasonable to suggest that CTs were not responding to hair deflection induced skin displacement, but rather to the hair deflection stimulus itself” (page 23).

Concerning the response firing rates, the data presented do not seem to be of typical physiological response rates for CTs to one of its preferred stimuli. All other CT papers show much higher firing rates for all other touch stimuli. If it really is the case that you find much lower firing frequencies, to hair deflection or stroking, this must be discussed.

A previous answer addresses this question: The low instantaneous frequency calculated was due to a calculation issue, where instantaneous frequency was being calculated from the push button onset/offset markers rather than between the first and last neural spike. After calculating the mean instantaneous frequency between the first and last spike during the stimulation period – the instantaneous frequency is now closer to the expected value of 21 spikes/second. In addition, with the same adjustment HFA brushing instantaneous frequency was recalculated as 31 spikes/second. The appropriate text and figures have been amended with the updated values. While we accept that the mean instantaneous frequency is still lower than previously published observations, the peak instantaneous frequencies (CT: M = 44 spikes/second; HFA: M = 70 spikes/second) are within the expected range (Löken et al., 2009, Nature Neuroscience). Therefore, focusing on the peak instantaneous frequency overcomes the issue of approximate stimulus timings, and indicates that all 15 CTs presented show a soft brush response that is within the typical range.

Specific points in the paper:

Abstract and key points

It says, "C-tactile afferents... have a functionally relevant anatomical relationship with hair follicles". This may be taking your message too far. You show that CTs can be activated via hair deflection and hair plucking, but this all relates more specifically to changes in skin displacement. You cannot be sure of whether it is truly hairs or just the skin, you did not look at the anatomical relationship and you did not chemically remove the hair. Although the hair plucking hints that CTs are coupled to hairs, there is still a big skin displacement, thus it seems like this is still an open question and it could just be a correlation between skin displacement and firing.

This is a valid point; indeed, skin and hair are structures that move dependently up to a certain extent. We believe that the main concern behind this comment is that deflecting a hair doesn't directly relate to the potential location of the C-tactile afferent, especially when compared with the impact of other stimulations such as normal skin indentations delivered with monofilaments. If

the C-tactile (CT) afferents were not located in close proximity to hair follicles but rather closer to the skin surface like other types of tactile mechanoreceptors, then when the deflected hair is near its maximum bending or being released—when the units fire the most (see Figure 3, blue bars for examples)—the skin surface deformation should be comparable to the mechanical stimuli used to assess the units' threshold (monofilaments between 0.4 to 4.0 mN). To test this hypothesis, we conducted an additional experiment to quantify the amount of surface skin displacement during monofilament indentations, hair deflection, and when the skin is at rest. This study showed no significant difference in surface skin displacement between the skin at rest (no stimulation) and during the transition of hair from maximal bending to its natural position. Additionally, the study revealed significant differences between hair deflection and monofilament indentations, supporting the hypothesis that CTs can be spatially closely associated with hair and hair follicles. The results section now includes figure 6: **“A pairwise comparison (Tukey's HSD test, p-value adjusted with a family-wise alpha of .05) was conducted to compare skin displacements among the different conditions (see Figure 6). Significant differences ($p < 0.05$) were found among all conditions except between the idle and hair deflection conditions ($p < 0.25$). The average displacement (in microns) for each condition was: idle at 2.04 (95% CI: 0.90, 3.17); hair deflection at 8.14 (95% CI: 5.48, 10.80); monofilament 0.4mN at 17.09 (95% CI: 13.63, 20.54); monofilament 0.7 mN at 25.28 (95% CI: 20.95, 29.60); monofilament 4.0 mN at 60.93 (95% CI: 56.39, 65.47); and monofilament 6.0 mN at 79.18 (95% CI: 71.70, 86.67). This suggests that all the different types of stimulation have a different impact on the skin surface and that bending one hair induced less surface skin deformation than a 0.4mN monofilament indentation. Furthermore, no significant difference was detected in surface skin displacement between the skin at rest (no stimulation) and during the transition of hair from maximal bending to its natural position. We have shown that any such deformation is significantly smaller than that caused by even the smallest monofilaments”** (page 19).

Introduction

In the introduction, there are a few papers that need to be included to introduce exactly what we know about the responses of CTs/C-LTMRs to hair deflection. In cats, Iggo (1960, J Physiol) showed that some C-LTMRs are sensitive to hair movement, but not others. In a follow-up study, Iggo & Kornhuber (1977, J Physiol) said that transmission of the deformation of the skin by hair displacement increased C-LTMR firing in cats (summarized). They even quantified that very small hair movements in the range of 10-20 μm could excite C-LTMRs. They also brought up the interesting point that afterdischarge could be directly related to hair displacement (and replacement), although hairs were not a sufficient factor to see this afterdischarge that still occurred on hair removal. Therefore, it is difficult to conclude the exact structural and functional relationship between CTs/C-LTMRs, as they are clearly co-related, but hairs are not sufficient for C-LTMR responses. Concerning human literature, three papers appear to mention CT responses to hair movement. Olausson et al (2010, Neurosci Biobeh Rev) stated that qualitative observations indicate that most CTs are not specifically sensitive to hair movements. Nordin (1990, J Physiol) found that, on the face, a single unit with scalp hairs arising from its receptive field responded vigorously to hair movements, but there was little or no response to sustained displacement of

hairs; however, this was just one CT and scalp hairs can be rather thick. Further, Ackerley et al (2014, J Neurosci) stated that CTs did not respond to air puffs, which activated hair afferents. Therefore, your work needs to be put into perspective in the introduction (and discussion) in relation to these previous findings that in animals, C-LTMRs sometimes elicited a response from hair deflection and in humans, there was little evidence for a response from CTs to hair movement.

We thank the reviewer for this detailed response. The introduction section now includes the suggested references to more clearly position the rationale and findings of the present study in relation to previous animal and human hair deflection studies. **“Movement of the hair shaft activates A β and A δ fibres in both the cat and rabbit, indicating a close proximity of the nerves to the hair follicle (Brown and Iggo, 1967; Burgess et al., 1968). In cats, Iggo (1960) also showed that some, but not all, C-LTMRs are sensitive to hair movement. Indeed, movements of both guard and down hairs as small as 10-20 μ m are sufficient to induce firing (Iggo & Kornhuber, 1977). It is suggested that small movements of hairs induced by epidermal movements due to restorative movements of the epidermal surface following skin displacement could contribute to after-discharges in C-LTMRs (Iggo & Kornhuber, 1977), due to restorative movements of the epidermal surface following skin displacement. Data on the responsiveness of CTs to hair movement in humans are sparse. Nordin (1990) described a single CT unit innervating human scalp hair that responds vigorously to displacement and replacement of a single hair, but not to sustained displacement”** (page 22).

This is also expanded upon in the discussion: **“Given that CTs can innervate more than one follicle within a receptive field, it could also explain the receptive field properties of CTs which typically exhibit several hotspots of mechanical sensitivity, which could potentially reflect the location of hair follicles (Wessberg et al., 2003). CTs remained sensitive to soft brushing and/or vonFrey monofilament indentation following the removal of individual hairs. However, it is not possible to know whether these were responses to direct skin stimulation or due to movement of other hairs within the receptive field. We did not assess if removal of hairs changed the receptive field properties (e.g., altered the number or sensitivity of the hotspots after shaving of the hair as described by Wessberg et al. (2003)”** (page 22).

In the middle of the introduction, when you write about hairs being present on the glabrous skin of the rodent paw (Walcher et al, 2018), it seems important to add that this is the hindpaw. The reference explicitly states that the hairs in glabrous mouse skin are very few and that they are only found in the hindpaw. Therefore, it is imprecise to make a point of mice having glabrous hair afferents when there really are very few and it is only on their hind feet.

This has been updated to clearly state that this is the hindpaw of the mouse: **“There are no hair follicles in glabrous skin in humans (McGlone et al., 2010) although a small number are present in the middle of the mouse hindpaw (Walcher et al., 2018)”** (page 4).

Methods - participants

Did the participants sign an informed consent form? Did it conform to national and international regulations for human experiments?

The methods sections now states that the study conformed with the Declaration of Helsinki and informed consent was obtained from all participants.

Methods - hair deflection

Can you give more precise information about how many repetitions of stimuli were given?

Phrasing has been amended to: **“Following multiple (between two and six) repetitions of hair deflection, the hair was pulled with increasing force until removed”** (page 8).

Methods - tracking hair deflection

This part is very interesting, yet there is little information on how this was done (e.g. camera, spatial and temporal resolution of videos, how an individual hair was actually tracked, software for tracking/analyzing). Further, the results of the tracking are not at all used and this would be extremely interesting to correlate the speed (and/or duration) of the bending to the firing of the units.

An additional methods subsection has been included describing hair deflection video acquisition and preprocessing, optical flow computation, computation of cumulative displacement fields, and visualisation and scaling.

Methods and results

You write about 'slow brush stroking', but this is vague. It is difficult to interpret the data when the actual speed, or at least an estimation of it, is not provided. This impacts on the results for firing frequency, which should change depending on stroking speed for all mechanoreceptors. Typically, any stroking from 0.3 to 30 cm/s would show mean firing rates of at least 20 spikes/s for CTs, whereas hairs would be from about 10 to 150 spikes/s (Löken et al, 2009; Ackerley et al, 2014). You state that slow stroking gave median firing rates of just 14 spikes/s for CTs and 37 spikes/s for HFAs, which is atypical for both. Did you track the stroking via video, like you tracked the hair movements or have some way of marking the speed of the stroking? If not, this is a variable that is not controlled and could have implications for the interpretation.

To clarify we have added a statement indicating that brushing was:

“delivered manually by a trained experimenter with a soft brush at a velocity of approximately 3cm/s” (page 7).

We are very grateful to the reviewers for finding this. Brush stroking velocity was not specifically recorded; however, it was approximately 3 cm/s. The low instantaneous frequency calculated for brushing of CT was due to a calculation issue, where instantaneous frequency was being calculated from the push button onset/offset markers rather than between the first and last neural spike. After measuring between the first spike and the last spike during the stimulation period – the instantaneous frequency was close to the expected value, 21 spikes/second. In addition, with the

same adjustment HFA brushing instantaneous frequency was recalculated as 31 spikes/second. This has been amended as appropriate in the text and figures. See also the response to reviewer 1 above.

Results

You mention in the results that you also tested air puff responses of CTs (which is not in the methods). As said above, CTs have not typically been shown to respond to air puffs, therefore this should be better explained. One thing that would again be useful to have is the timing, especially with regards to the analysis of firing frequency. It seems that if you are unsure of stimulus timings, peak frequency could be a good option to focus on and/or the response in the first second only (as long as the stimulus was longer than a second).

Added statement in the methods describing the air puff device: **“A manual squeezable air puff device was used to deliver a weak but focused jet of air over the receptive field”** (page 6). In the context of previous studies reporting that CTs do not respond to air puff, it remains unclear why we observed CTs firing in response to air puff, however this response was tested systematically.

We have commented further on the differences between the previous literature and our more systematic assessment of CT responses to hair movement in the discussion (see below).

Results - hair plucking

You say that you tested 3 HFAs to hair plucking, but do not give any further information. Could you also add the same data as per the CTs, i.e. the instantaneous frequency data, and ideally add an example to Fig. 4?

We have added an HFA exemplar plot to figure 4 showing an absence of after-discharge following hair plucking. This has been reported as: **“HFA units (n = 2) did not exhibit after-discharge following hair plucking (mean instantaneous frequency = 0 mean peak instantaneous frequency = 0)”** (page 16).

Figures - overall

The figures would benefit from better scale bars. For example, the horizontal scales for spike trains could have time markers or at least the scale bar at the beginning of the spike firing, as would be more typical. Also, the vertical scales often only consist of one or two points, with no line indicating the axis. A more indicative scale would be much more informative, especially considering the question over the firing rates.

Thank you – vertical scales in figures have now been changed from log10 to linear scales and more detailed ‘tick marks’ have been added to all scales to improve readability. Also, horizontal scale bars to show stimulus timings have been added.

Figures - Fig. 3

Why is the vertical scale in Fig. 3 log₁₀? This seems quite difficult to interpret, especially in comparison to the literature. I guess it is for the comparison between CTs and HFAs, but it looks like the results would be even clearer on a linear scale. Again, it seems strange that CTs would produce such low mean firing frequencies to stroking.

Thank you, the vertical scale has now been changed from log₁₀ to a linear scale that is now more intuitively interpreted. After measuring between the first spike and the last spike during the stimulation period – the instantaneous frequency is calculated much closer to expected, 21 spikes/second.

Figures - Fig. 5

Could you add the actual CT recording to Fig. 5?

Added raw neural traces to figure 5, which more clearly presents the CT neural response during piloerection.

Discussion

In the part about monofilaments and CT receptive fields in relation to hairs, you need to develop these points further. You say the 'mechanical detection threshold of CTs using monofilaments', but this must not be confused with perception, so I suggest to use a better term, which could be the 'force activation threshold'.

'Mechanical detection threshold' has been amended to 'force activation threshold' as suggested.

As the hairs appear to act as a support for mechanoreceptors in hairy skin, it is no surprise that hair deflection could activate any mechanoreceptor, thus this point about monofilaments would surely be the same for measuring thresholds for all mechanoreceptors.

We have now added more data regarding skin deformation during hair deflection. With respect to the comment that if a hair deflection causes significant deformation on the skin this could activate any low threshold mechanoreceptor, this is not the case as field units, for example, do not respond to hair deflection despite a likely anatomical proximity with hair follicles. The following has been added to the discussion to highlight this. **"Comparison with field units is interesting in this respect. Field units in humans have exquisite mechanical sensitivity (Nagi et al. 2019) and do not respond to hair deflection. Although the peripheral endings of field units in humans are yet to be defined, field afferents in mice, which have expansive receptive fields, are intimately related to hair follicles yet they do not fire in response to hair deflection (Bai et al 2015). This**

indicates that having an intimate anatomical arrangement with hair follicles does not necessarily imply that these endings fire in response to hair deflection” (page 23).

Also, CT force activation thresholds are so low (e.g. often well less than 5 mN), would moving a hair at this force really have an effect on CT firing?

We thank the reviewer for this comment. We refer to the papers by Iggo (1960) and Iggo & Kornhuber (1977) that the reviewer kindly suggested that we mention in the introduction which documented that

“C-hair receptors were excited by moving only the tips of hairs, and the rate of discharge in response to such mild stimulation was surprisingly high; e.g. for one unit the discharge was at 25/sec when the tips of the hairs were brushed and was not more than doubled by a firm stroke of the skin. Another unit gave a burst of impulses lasting 1:5 sec, with a peak frequency of 32/sec, when the hairs were bent backwards”.

and

“The sensitive C-mechanoreceptors can be excited by small (10-20 um) movements of both guard and down hairs. With the stimulus probe attached to one guard hair and several down hairs, a small movement (45 um) was sufficient to cause the discharge of 5 impulses in 1 sec.”

Any vonFrey monofilament used during our microneurography recordings would have the potential to move a hair to the extent of 10-20 um. We also refer to the classification of A-beta hair follicle afferents (Vallbo et al (1995)), which are defined by response to hair deflection (air puff, manual hair deflection) rather than punctate stimuli. Since you can catch hairs with the vonFrey monofilament and produce spiking, thresholds are unreliable. In fact, because we show that CT afferents respond to hair deflection, we made the argument that testing of their mechanical thresholds is problematic for the same reason.

Further, what is your point about the receptive field in relation to the Wessberg et al (2003) paper, why would your results explain the maps of 1-9 hot spots?

Thank you for this comment. Our reasoning for our point about the receptive fields with 1-9 hot spots is that the hotspots could reflect the location of hair follicles within the receptive field. We have shown (data now added) that CTs can respond to the deflection of more than one single hair within the receptive field (see above) which could account for several hotspots per fibre. We have expanded upon this in the discussion.

“Given that CTs can innervate more than one follicle within a receptive field, it could also explain the receptive field properties of CTs which typically exhibit several hotspots of mechanical sensitivity, which could potentially reflect the location of hair follicles” (page 22).

Again, I am not convinced that the gentle bending of a hair only could really generate the responses seen, although it could contribute.

We thank the reviewer for this comment. Our findings in humans are similar to that of Iggo (1960) and Iggo & Kornhuber (1977), as described above. We have expanded upon this point further in the discussion where we demonstrate with the use of optical coherence tomography that skin movement during fine hair movements was not significantly greater than at a resting state, suggesting that CTs respond to hair deflection itself, with minimal skin deformation. This was further confirmed using cumulative flow vector analysis that showed no substantial skin deformation around the stimulated hair during bending.

The receptive fields detailed in the Wessberg paper would be expected from mechanoreceptors and C-fibers; would you predict something different on hair removal? This is a point that could be followed up on.

Thank you for this comment. We have made further comment about firing of CTs following hair removal in the discussion in relation to hair removal.

“CTs remained sensitive to soft brushing and/or vonFrey monofilament indentation following the removal of individual hairs. However, it is not possible to know whether these were responses to direct skin stimulation or due to movement of other hairs within the receptive field. We did not assess if removal of hairs changed the receptive field properties, e.g., altered the number or sensitivity of the hotspots described by Wessberg et al 2003” (page 22).

In the paragraph starting, "It may be suggested that hair deflection...", I am not sure what you point is about skin deformation and other hairs touching is, and how HFAs have the same caveat. Even if you move a single hair, it will likely cause the underlying skin to move.

Thank you for your comment. We were exploring potential causes (other than hair movement per se) that could explain the findings.

- 3) For example, if while moving one hair we caused another hair crossing the path of movement to deflect and physically touch the skin this could potentially activate a low threshold unit to be activated by direct skin contact.
- 3) Nevertheless, we appreciate that hair deflection may cause movements within the follicle, causing CT firing. An alternative explanation is chemical transmission as suggested by Agramunt (2023), showing that outer root sheath cells (within the hair follicle) release ATP and the neurotransmitters serotonin and histamine in response to mechanical stimulation. We have expanded upon these possibilities in the discussion.
- 3) A-beta hair follicle afferents are defined by their responsiveness to hair deflection in microneurography recordings. We know that humans have large, myelinated endings associated with hair follicles (HFA) and it is reasonably assumed that their firing is related to the hair deflection rather than hairs touching the epidermis during movement etc (unlike field units, which are exquisitely mechanically sensitive but do not respond to hair deflection but still have an association to hair follicles (in mice; Bai et al, 2015, Cell).

The following has been added to the discussion:

“Our findings suggest there is a close functional anatomical relationship between CT afferents and hair follicles, but there are several possible mechanisms by which hair deflection could induce firing. It could be argued that during the process of hair deflection other hairs were also moved causing a brushing against the skin surface. However, hair deflecting stimuli were delivered under magnification and great care was taken to avoid either direct or indirect mechanical stimulation of the skin surface. Therefore, technical factors like this seem very unlikely. One possibility is that hair movement causes deformation of the peri-follicular skin, and it is this that initiates firing of CTs. Only two (of fifteen) CTs had a mechanical threshold as low as 0.4mN, which our optical coherence tomography experiment confirmed produces 17µm skin displacement. Given that hair deflection induced skin displacement (8µm) was significantly less than this (8µm - approximately half the displacement for that of a 0.4mN monofilament), it is reasonable to suggest that CTs were not responding to hair deflection induced skin displacement, but rather to the hair deflection stimulus itself. However, it remains possible that peri-follicular skin movement, when present, may also contribute. Comparison with field units is interesting in this respect. Field units in humans have exquisite mechanical sensitivity (Nagi et al., 2019) and do not respond to hair deflection. Although the peripheral endings of field units in humans are yet to be defined, field afferents in mice, which have expansive receptive fields, are intimately related to hair follicles yet they do not fire in response to hair deflection (Bai et al., 2015). This indicates that having an intimate anatomical arrangement with hair follicles does not necessarily imply that these endings fire in response to hair deflection.

Hair movements and bending will cause mechanical strain within the follicle. To induce firing of CT afferents this would require the peripheral endings of CT afferents to be tightly linked anatomically to the follicle, like the relationship between C-LTMRs and awl/auchene and zigzag hairs in mice (Li et al., 2011). Given that skin deformation induced by hair deflection is minimal, and less than that induced by low force Von Frey monofilaments, an anatomical connection to the hair follicle appears likely. In either case the strong expression of Piezo2 in CTs/human C-LTMRs documented by single cell transcriptomics from human dorsal root ganglia (Yu et al., 2024) will facilitate this mechanosensitivity. A further possibility is that chemical transmission could contribute to afferent firing during hair deflection. It has been shown that outer root sheath cells in human hair follicles can release serotonin and histamine and that these transmitters can activate co-cultured with murine sensory neurons, detected by calcium imaging. While sensory neuron sub-types were not defined, and whether the same would occur in human neurons is unknown, it does suggest the viability of a chemical signaling mechanism in addition to a mechanism mediated by mechanosensitive ion channels such as Piezo2 (Agramunt et al., 2023)” (page 23-24).

Overall, the discussion does raise some interesting points, but it could be shorted in some parts to include a number of important things have not been considered, e.g.:

- Why hair deflection would be such a pertinent stimulus for CTs

This has been added to the discussion section (see above comment)

We suggest that hair deflection is a pertinent stimulus for CTs due to a close functional anatomical relationship with the hair follicle: **“Hair movements will cause mechanical strain within the follicle. To induce firing of CT afferents this would require the peripheral endings of CT afferents to be tightly linked anatomically to the follicle, like the relationship between C-LTMRs and awl/auchene and zigzag hairs in mice (Li et al., 2011). Given that skin deformation induced by hair deflection can be minimal, and less than that induced by low force vonFrey monofilaments, an anatomical connection to the hair follicle appears likely. The strong expression of Piezo2 in CTs/human C-LTMRs documented by single cell transcriptomics from human dorsal root ganglia (Yu et al 2024) will facilitate this mechanosensitivity”** (page 24).

- Why your results are different to literature that indicates some sensitivity of C-LTMRs to hair deflection and little sensitivity in CTs. Also, it is not clear if the sample you present is true for all CTs, i.e. whether you only document CTs that have a positive hair deflection response or whether you see CTs that are unresponsive.

All 15 of the CTs presented here responded to hair deflection, however we are not suggesting that 100% of CTs in humans respond based on this small sample. We have add to discussion section: **“Data on the responsiveness of CTs to hair movement in humans are sparse, and we can only speculate on why our results indicate a greater CT sensitivity to hair deflection in comparison to previous literature. We undertook a rigorous systematic exploration of 15 consecutively recorded CTs where a single hair was carefully deflected under magnification, and perhaps previous studies have not focused on hair deflection responses in detail. Nordin (1990) described a single CT unit innervating human scalp hair that responds vigorously to displacement and replacement of a single hair, but not to sustained displacement. Ackerley et al., (2014, J Neurosci) stated that CTs do not respond to air puffs, which activated A β hair follicle afferents. Given that CTs can innervate more than one follicle within a receptive field, it could also explain the receptive field properties of CTs which typically exhibit several hotspots of mechanical sensitivity, which could potentially reflect the location of hair follicles (Wessberg et al., 2003). CTs remained sensitive to soft brushing and/or vonFrey monofilament indentation following the removal of individual hairs. However, it is not possible to know whether these were responses to direct skin stimulation or due to movement of other hairs within the receptive field. We did not assess if removal of hairs changed the receptive field properties (e.g., altered the number or sensitivity of the hotspots after shaving of the hair as described by Wessberg et al., (2003). To what extent CTs/C-LTMRs have a role in the perception of individual hair deflections is unknown”** (page 22).

- Why bending a hair gives a much stronger response in CTs than the actual HFA.

Given that HFAs preferentially encode hair movement, it is unclear why neural responses to hair deflection were greater in CTs compared to HFAs. We deflected hairs cautiously with a relatively slow velocity ensuring not to move adjacent hairs or touch the skin, it seems likely that HFAs respond more vigorously to rapid bending of the hair since they generally respond better to rapid skin stimulation (Loken et al, 2009).

- That there are inherent difference between the thickness of hairs (fur) between humans and other animals, which could produce different results.

We have added to the discussion section: **“A direct comparison of hair deflection responses is difficult due to differences between human hair and animal fur”** (page 21).

- There is also the point that the association between CTs and hair could be true for all mechanoreceptors in hairy skin (cf. Li et al, 2011), where the hairs provide an anchoring matrix structure, but taking it further, it does not mean that mechanoreceptors require hairs to function.

If human anatomy is consistent with that of the mouse (Li et al, 2021), each hair follicle type may provide anchoring structural support for a combination of cutaneous afferents. The field afferent is intimately related to the hair follicle, yet they do not fire in response to hair deflection (Bai et al 2015). This indicates that an intimate anatomical arrangement with hair follicles does not necessarily imply an inherent function, such as encoding of hair movement. At least a single hair innervated the receptive field of all 15 CTs that we recorded, however, we are not suggesting based on this that *all* CTs innervate hair follicles or that CTs require hairs to function, rather that the anatomy of the CT may mimic that of other mammals.

- A discussion of further experiments, e.g. speed of hair bending and if this relates to the typical CT inverted-U curve, chemical hair removal.

As suggested, the following has been added to the future directions section: **“To further understand the involvement of hairs within the receptive field of CTs responses should be compared before and after hair removal. Recording CT response to hair deflection of different velocities would further improve our understanding of how hair movement contributes to CT responses”** (page 25).

Minor comments

- Firing frequency is stated in Hz, but it is typically in spikes or impulses per second for physiology. Units of instantaneous frequency have been changed from ‘Hz’ to ‘spikes/second’.

- In the results, you write a few times things like 'mean frequency: Mdn = ...'. Do you mean the group median of the mean frequency per unit? This does not seem to make much sense. You could simply just present the group means.

All median scores have been recalculated to mean scores.

- Please add page and/or numbers to your paper to help follow with the review.

Page numbers have been added.

Dear Dr Marshall,

Re: JP-RP-2025-287706R1 " **Robust Coupling Between the C-Tactile Afferent and the Hair Follicle in Humans**" by Warren Moore, Johan Nikesjö, Otmane Bouchatta, Adarsh D Makdani, Pierre Hakizimana, Mikael Rousson, Basil Duvernoy, Sarah McIntyre, Laura Johanna Pehkonen, Anders Fridberger, Francis McGlone, Hakan Olausson, Saad S Nagi, and Andrew Marshall

Thank you for submitting your manuscript to The Journal of Physiology. It has been assessed by a Reviewing Editor and by 2 expert referees and we are pleased to tell you that it is acceptable for publication following satisfactory revision.

REVISION CHECKLIST:

We look forward to receiving your revised submission.

Yours sincerely,

Nathan Schoppa
Senior Editor
The Journal of Physiology

REQUIRED ITEMS

- The contact information for the person responsible for 'Research Governance' at your institution needs to be provided. This includes their name and an institutional email address. Please ensure the contact is not an author on this paper and provide an alternate contact if necessary, or confirm in the submission form that the author whose email was provided has sole responsibility for research governance. This is the person who is responsible for regulations, principles and standards of good practice in research carried out at the institution, for instance the ethical treatment of animals, the keeping of proper experimental records or the reporting of results.

EDITOR COMMENTS

Reviewing Editor:

Thank you for submitting your revised manuscript to The Journal of Physiology. The two independent reviewers are satisfied with your thorough revisions and inclusion of additional experiments. However, there are a few inconsistencies noted by Reviewer 2 in the Clean and Tracked versions of the submitted manuscript. Please check and resubmit.

Senior Editor:

Congratulations! The two original external referees were quite satisfied with the changes that you made in the revised manuscript and the work is now deemed acceptable for publication, provisional on the authors correcting a few inconsistencies in the presentation. These will need to be addressed in another revision.

REFeree COMMENTS

Referee #1:

I would like to thank the authors for their thorough and meticulous responses to all questions and comments. I highly value the addition of optical coherence tomography data and the analyses performed to strengthen the findings.

The manuscript now includes required experimental details and comprehensively discusses these findings within the context of published literature.

This outstanding paper will have a significant impact on the research field related to these somewhat enigmatic CT afferents in humans. I anticipate that this paper will be met with exceptional interest by the journal readership, initiating further research to test the generalisability of these findings and provide detailed answers to still remaining questions.

Referee #2:

I thank the authors for their detailed replies to my comments and the modifications made much improve the clarity of the paper. The methods and analysis are especially clearer now. The new section on hair deflection tracking is a very nice addition to the paper. I have just a couple of minor comments below. Altogether, this provides the basis of a strong and original research paper that contributes to the literature.

There seem to be inconsistencies between the submitted tracked-changes version of the paper and the clean version. I found the below two parts in the methods, but please do check for any other differences.

The section 'Microneurography unit classification' is different between the track-changes version and the clean version. Please can you check this and made sure to include, 'and had a mechanical threshold below of 4mN or below, defined as at least a single spike response on at least 50% of occasions to monofilament indentation'.

In the section 'Microneurography Hair Deflection', the text is also different between the two submitted versions. Please add 'A minimum of two spikes was chosen to provide an extra layer of confidence that the response seen was an evoked one and also because this is needed to detect an instantaneous firing frequency' to the clean version.

The authors have not fully revised their paper with respect to my previous minor comment, 'Firing frequency is stated in Hz, but it is typically in spikes or impulses per second for physiology'. Although it is changed in the text, the figure y-axes would be better as spikes/s.

END OF COMMENTS

Rebuttal

We would like to thank the reviewers for their positivity towards the manuscript and their attention to detail.

There seem to be inconsistencies between the submitted tracked-changes version of the paper and the clean version. I found the below two parts in the methods, but please do check for any other differences.

All discrepancies between the clean version and tracked changes version have now been corrected.

The section 'Microneurography unit classification' is different between the track-changes version and the clean version. Please can you check this and made sure to include, 'and had a mechanical threshold below of 4mN or below, defined as at least a single spike response on at least 50% of occasions to monofilament indentation'.

Thank you for spotting this – we have updated in the clean version to be consistent with the tracked changes.

In the section 'Microneurography Hair Deflection', the text is also different between the two submitted versions. Please add 'A minimum of two spikes was chosen to provide an extra layer of confidence that the response seen was an evoked one and also because this is needed to detect an instantaneous firing frequency' to the clean version.

Thank you for spotting this – we have updated in the clean version to be consistent with the tracked changes.

The authors have not fully revised their paper with respect to my previous minor comment, 'Firing frequency is stated in Hz, but it is typically in spikes or impulses per second for physiology'. Although it is changed in the text, the figure y-axes would be better as spikes/s.

All figures have been reproduced with 'spieks/s' in place of 'Hz' to be consistent with the text throughout.

Dear Dr Marshall,

Re: JP-RP-2025-287706R2 " **Robust Coupling Between the C-Tactile Afferent and the Hair Follicle in Humans**" by Warren Moore, Johan Nikesjö, Otmane Bouchatta, Adarsh D Makdani, Pierre Hakizimana, Mikael Rousson, Basil Duvernoy, Sarah McIntyre, Laura J Pehkonen, Anders Fridberger, Francis McGlone, Hakan Olausson, Saad S Nagi, and Andrew Marshall

We are pleased to tell you that your paper has been accepted for publication in The Journal of Physiology.

Yours sincerely,

Nathan Schoppa
Senior Editor
The Journal of Physiology

If you would like to receive our 'Research Roundup', a monthly newsletter highlighting the cutting-edge research published in The Physiological Society's family of journals (The Journal of Physiology, Experimental Physiology, Physiological Reports, The Journal of Nutritional Physiology and The Journal of Precision Medicine: Health and Disease), please click this link, fill in your name and email address and select 'Research Roundup':
<https://www.physoc.org/journals-and-media/membernews>

- You can help your research get the attention it deserves! Check out Wiley's free Promotion Guide for best-practice recommendations for promoting your work at: www.wileyauthors.com/eeo/guide. You can learn more about Wiley Editing Services which offers professional video, design, and writing services to create shareable video abstracts, infographics, conference posters, lay summaries, and research news stories for your research at: www.wileyauthors.com/eeo/promotion.

EDITOR COMMENTS

Reviewing Editor:

Thank you for attending to the remaining concerns raised by Reviewer 2. I am satisfied these have been adequately addressed.

Senior Editor:

Thank you for submitting your latest revision. Congratulations on its acceptance!